# Transcriptomic decoding of surface-based imaging phenotypes and its application to pharmacotranscriptomics

Christine Ecker[1,2,3] ✉, Charlotte M. Pretzsch[2], Johanna Leyhausen[1,3,4], Lisa M. Berg[1], Caroline Gurr[1,3], Hanna Seelemeyer[1,3], Grainne McAlonan[2], Nicolaas A. Puts[2], Eva Loth[2], Flavio Dell'Aqua[2], Luke Mason[2], Tony Charman[5], Bethany Oakley[2], Thomas Bourgeron[6], Christian Beckmann[7], Jan K. Buitelaar[7], Celso Arango[8], Tobias Banaschewski[9], Andreas G. Chiocchetti[1], Christine M. Freitag[1], Elke Hattingen[10], Dilja Krueger-Burg[11], Michael J. Schmeisser[11], Jonathan Repple[3,12,13,14], Andreas Reif[12,13] & Declan G. Murphy[2]

Imaging transcriptomics has become a power tool for linking imaging-derived phenotypes (IDPs) to genomic mechanisms. Yet, its potential for guiding CNS drug discovery remains underexplored. Here, utilizing spatially-dense representations of the human brain transcriptome, we present an analytical framework for the transcriptomic decoding of high-resolution surface-based neuroimaging patterns, and for linking IDPs to the transcriptomic landscape of complex neurotransmission systems in vivo. Leveraging publicly available Positron Emission Tomography (PET) data, we initially validated our approach against molecular targets with a high correspondence between gene expression and protein binding. Subsequently, we used the cortical gene expression profiles of candidate genes to dissect two discrete classes of GABA$_A$-receptor subunits, each characterized by a distinct cortical expression pattern, and to link these to specific behavioural symptoms and traits. Our approach thus represents a future avenue for in vivo pharmacotranscriptomics that may guide the development of targeted pharmacotherapies and personalized interventions.

Imaging transcriptomics, the study of correlations between gene-expression patterns and spatially varying properties of brain structure and function[1], has become a powerful tool for exploring the putative molecular underpinnings of neurotypical and neurodiverse brain organization. Here, large open-access repositories featuring genome-wide expression profiles sampled across the brain, e.g., the Allen Human Brain Atlas (AHBA[2]), are used to identify genes with an expression signature that spatially aligns with a structural or functional imaging phenotype. These gene sets are then functionally annotated or tested for an enrichment of candidate gene sets and/or genetic pathways[3]. In so-called virtual histology studies, for example, imaging-derived phenotypes (IDPs) are tested for an enrichment of cell-type specific genes to probe their cellular composition[4]. Other studies have examined genes within co-expression modules underpinning typical brain development to explore the role of transcriptional developmental programs on IDPs in neurodevelopmental conditions (e.g[5].). Imaging transcriptomics thus represents a promising avenue for bridging the gap between molecular mechanisms and macroscopic patterns of brain organization.

With the release of spatially-dense (i.e., vertex-level) cartographic representations of the human brain transcriptome (e.g.[6,7],), imaging transcriptomics also holds promise for dissecting the cortical architecture of complex neurotransmission systems and neuromodulatory pathways[8]. While there is often no 1:1 relationship between gene expression and receptor density[9], evidence suggests that patterns of regional variations in gene expression can provide important insights into the functional role of a molecular target (e.g[10,11].). For example, genes encoding γ-aminobutyric acid receptor (GABA_AR) isoforms have been reported to be differentially expressed across specific brain regions[12], which may explain their distinct behavioral effects when targeted pharmacologically. Specifically, isoforms containing the $\alpha_2$ subunit appear to be predominantly expressed in limbic brain regions and exhibit anxiolytic effects when targeted (reviewed in[13]. Conversely, isoforms containing the $\alpha_1$ subunit are expressed throughout the cortex, and have been shown to have primarily sedative, amnesic, and anticonvulsive effects[13]. This implies that the cortical expression profiles of specific molecular targets may reflect their functional significance, which could be investigated by spatially aligning (i.e., correlating) IDPs with candidate gene expression patterns. However, assessing spatial correlations between embedded high-resolution signals – such as associating IDPs with genome-wide expression patterns (i.e., transcriptomic decoding) or linking a large number of IDPs to a target pattern - remains a significant computational and statistical challenge[3].

In the current study, we therefore explored different strategies for linking high-resolution surface-based IDPs to gene expression patterns using spatially-dense representations of the human brain transcriptome generated from the AHBA[2]. These approaches were initially validated against publicly available Positron Emission Tomography (PET) atlas data of the human serotonergic (5-HT) system, which is known to strongly correlate with the mRNA expression levels of respective serotonergic genes in cortical brain tissue[14,15]. Subsequently, we utilized the vertex-level gene expression signatures to (i) decompose the complex transcriptomic landscape of genes encoding different GABA_AR isoforms, and (ii) explore their functional role by aligning their gene expression profiles with neuroanatomical imaging data from $N = 279$ children, adolescents and adults exhibiting varying levels of affective symptoms, such as anxiety and depression. The individuals' imaging phenotypes were characterized based on measures of cortical thickness (CT), a morphometric feature demonstrated to be modulated by variability in both genes and gene expression[16,17].

## Results

### Comparison of decoding techniques using PET atlas data of the serotonergic (5-HT) system

To link high-resolution IDPs with gene expression patterns, we initially evaluated multiple strategies for the statistical assessment of spatial correlations between surface-based imaging patterns and genome-wide expression signatures, subsequently referred to as gene expression decoding. All techniques are described in detail in the Methods section. In brief, these included: (i) a vertex-level approach using spatial autocorrelation ($\alpha$) preserving null models[18] of large-scale co-expression gradients, (ii) a Linear Mixed Effects (LME) model[19] (Supplementary Data Fig. 1a), and (iii) General Least Squares (GLS) decoding (Supplementary Data Fig. 1b). Notably, for the vertex-level decoding approach, we generated spatially-dense gene expression signatures across the cortical surface (Fig. 1), which were subsequently decomposed into a smaller set of co-expression gradients, for which spatial null models were generated (Fig. 2) (also see[6,18,20]). The sensitivity and specificity of the different techniques were assessed using a high-resolution in vivo PET atlas of the human serotonergic (5-HT) system[15], which included the serotonergic receptors 5-HT_1AR, 5-HT_2AR, and 5-HT_4R.

For these molecular targets, all approaches reliably detected a significant association between mRNA expression levels and PET protein binding, based on a nominal (uncorrected) $p_{perm}$-value < 0.05 (two-tailed). Consistent with previous findings[15], the strongest spatial correlation between protein binding and mRNA expression was observed for the 5-HT_1AR gene (HTR1A), followed by the 5-HT_2AR gene (HTR2A) and the 5-HT_4R gene (HTR4) (Fig. 3a, Supplementary Data Figs. 2, 3a). All techniques also accurately identified the target gene within the gene background based on $p_{adj} < 0.05$, except for the gradient-based decoding of HTR4, which did not reach genome-wide statistical significance. The different techniques also resulted in a similar relative ranking of individual genes with respect to the target

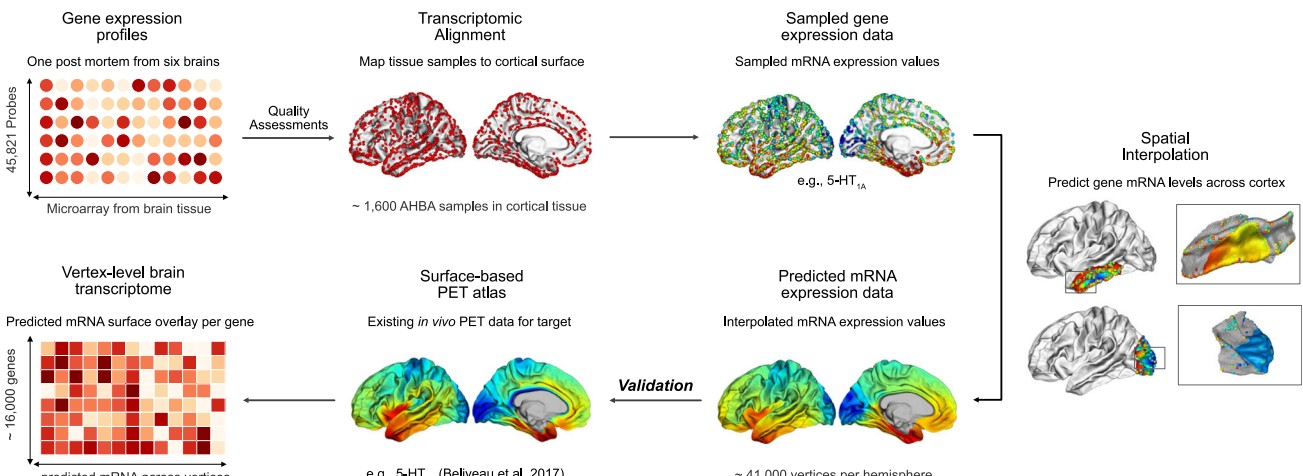

**Fig. 1 | Workflow diagram for generating high-resolution surface-based transcriptomic signatures for all cortically expressed genes with reliable mRNA microarray data in the Allen Human Brain Atlas (AHBA).** The AHBA data was initially quality assessed and normalized across genes and donors using the abagen toolbox[32]. Samples were then mirrored across hemispheres and assigned to the FreeSurfer fsaverage6 standard-space surface template (~41,000 vertices) via mesh representations of the donor brains. To derive spatially-dense (i.e., vertex-level) gene expression profiles across the entire cortical surface, the mRNA expression patterns were spatially interpolated using Gaussian Process Regression (i.e., ordinary Kriging), predicting mRNA levels for vertices without AHBA representations using existing samples within a geodesic vertex neighborhood of 40 mm. Spatial interpolation was performed for each of 15,633 quality assessed genes, resulting in a spatially-smooth vertex-level representation of the Human brain transcriptome. The figure shows the predicted transcriptomic signature of the serotonin (5-HT) 1 A receptor (5-HT1A), which was subsequently compared to available PET atlas data of the same molecular target (see[15]).

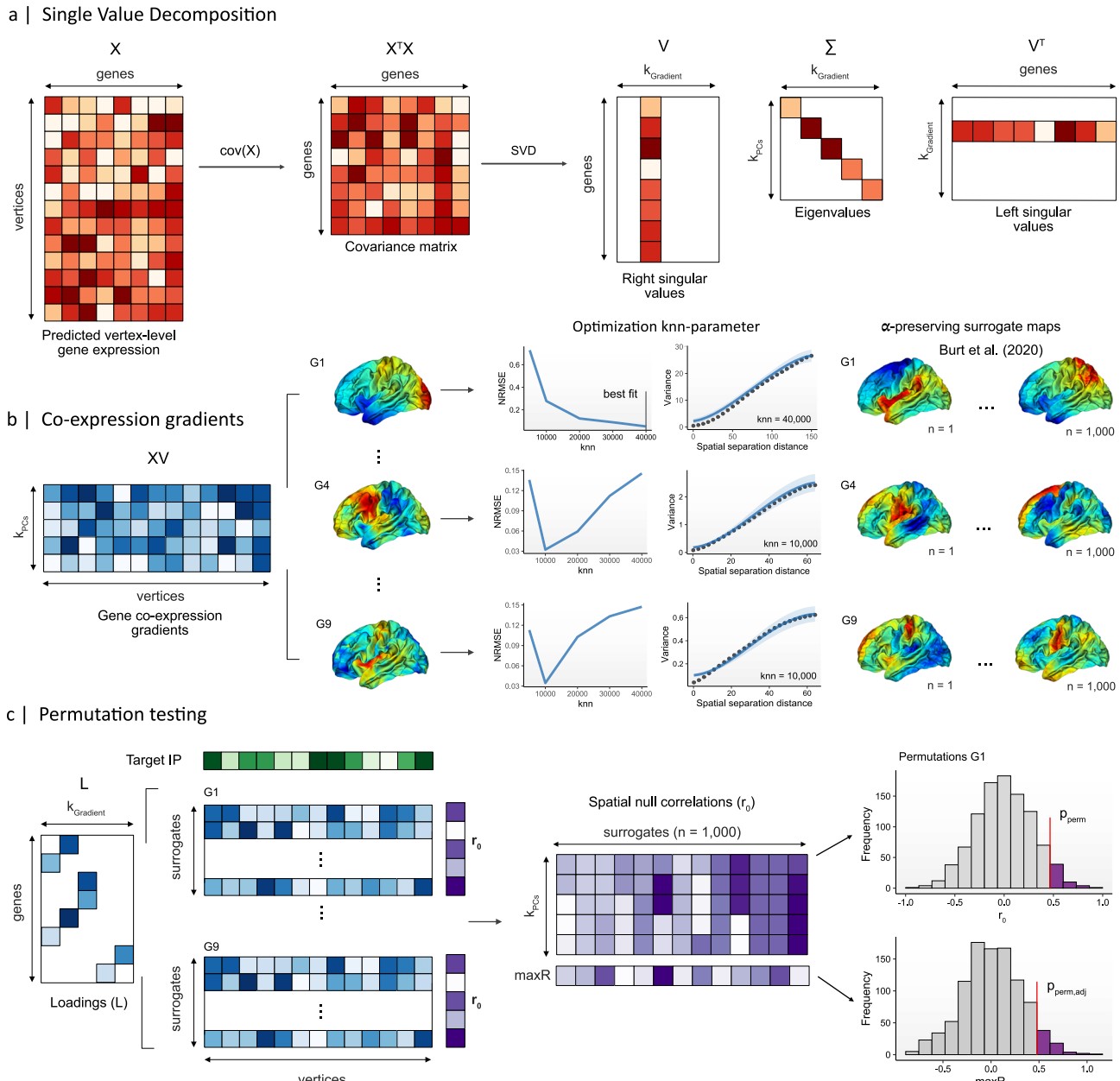

**Fig. 2 | Transcriptomic decoding of high-resolution surface-based imaging-derived phenotypes (IDPs). a** Singular Value Decomposition (SVD) was employed to reduce the spatially-interpolated mRNA expression signatures of all $N = 15{,}633$ genes in cortical brain tissue to a smaller subset of nine spatially-dense co-expression gradients (G1 to G9), which together captured ~41% of total variability in gene expression across the surface (see Supplementary Data Fig. 13a,b). **b** A total of $N = 1000$ spatial autocorrelation (α)-preserving null models (so-called surrogates[18]) were generated for each gradient pattern using the optimal k-nearest-neighbor (knn) parameter for each pattern (also see Supplementary Data Fig. 13c). These surrogate maps exhibit a similar degree of smoothness as the original patterns and were employed to testing the hypothesis of a significant spatial association

between a target pattern and the predicted cortical expression signature of a gene. Data are presented as mean values +/- standard deviation (across $N = 100$ surrogate fits). **c** Genes were allocated to co-expression gradients based on their maximum absolute loadings (L) on gradient (G) patterns (G1 to G9). For each observed spatial correlation between a target IDP and a gene's cortical expression profile, a non-parametric α-corrected p-value ($p_{perm}$, two-tailed) was identified based on the distribution of spatial null correlations ($r_0$) with the respective gradient pattern. To correct for multiple comparisons, $p_{perm}$-values were adjusted for ($p_{perm,adj}$) based on the empirical cumulative density function of the extreme value distribution (i.e., maximal spatial correlation or maxR) across gradient null-models (also see MaxT algorithm[38]).

---

pattern, as evidenced by significant Spearman Rank correlations between gene weights (Fig. 3c, Supplementary Data Figs. 2, 3c).

Overall, the gradient-based approach provided the best trade-off between sensitivity and specificity across various levels of statistical stringency, identifying a reasonable number of significant genes suitable for downstream enrichment analysis (between 100 and 2000 at $p_{adj} < 0.001$) (Fig. 3b, Supplementary Data Figs. 2, 3b). In comparison,

LME-decoding identified the largest number of significant transcriptomic associations at $p_{adj} < 0.05$. Yet, this number decreased rapidly when more conservative p-value thresholds were applied (Fig. 3b, Supplementary Data Figs. 2, 3b). Thus, although LME-decoding has substantial exploratory potential for detecting transcriptomic associations, it is also prone to generating false positives, likely due to spatial autocorrelations within the embedded

### a | 5-HT$_{1A}$R receptor density and mRNA profiles

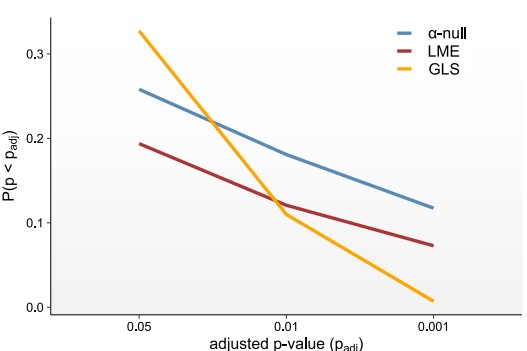

$r_{spatial}$ = 0.852
$p_{perm}$ < 0.001
$p_{perm.adj}$ <0.001

### b | Probability of significance

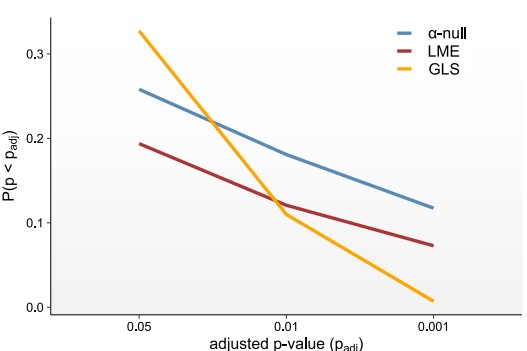

### c | Rank correlations between gene weights

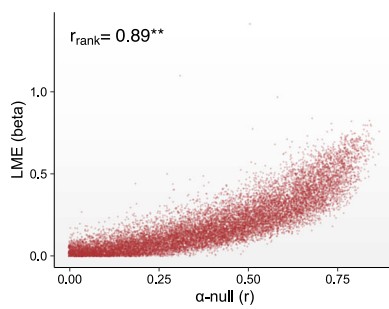
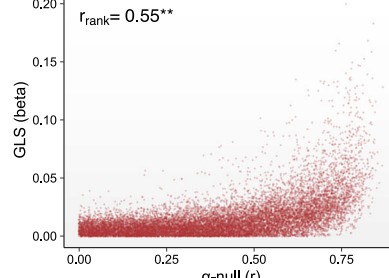
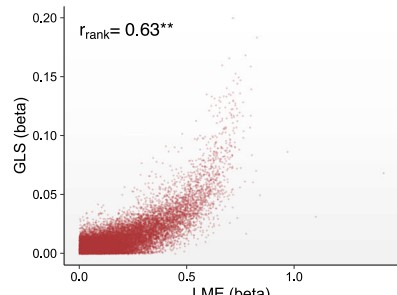

### d | Decoded gene set sizes

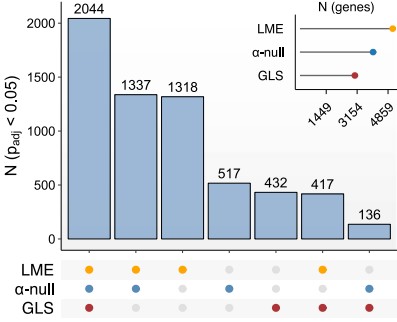
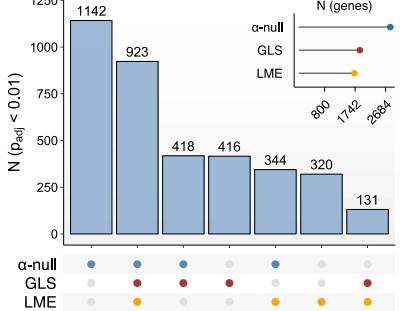
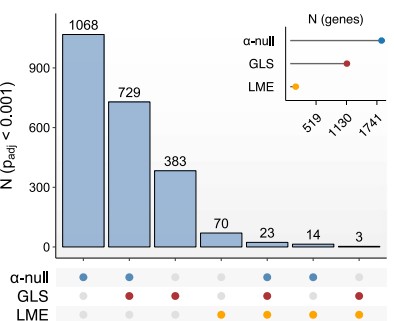

**Fig. 3 | Surface-based transcriptomic decoding of the serotonergic 1A receptor (5-HT$_{1A}$R). a** Spatial correlations between the average receptor density (B$_{max}$) of the 5-HT$_{1A}$R based on the high-resolution in vivo Positron Emission Tomography (PET) atlas of the human serotonergic system provided by ref. 15 (upper panel) and the predicted mRNA expression profile of the *HTR1A* gene (lower panel). **b** False positive rate (FPR) of the α-null (blue), Linear Mixed Effects (LME, red), and General Least Squares (GLS, yellow) decoding techniques across different adjusted *p*-value thresholds (p$_{adj}$). For LME- and GLS-decoding, *p*-values were adjusted for multiple comparisons using False Discovery Rate (FDR) adjustments. For α-null decoding, adjusted permutation *p*-values (p$_{perm,adj}$) were used. **c** Spearman rank correlations between the gene weights across approaches. Gene weights were determined by the slope of the regression line (i.e., beta) for the LME- and GLS-decoding approach, and by the spatial Pearson correlation between receptor density and gene expression maps for the α-null decoding approach. ** *p*-value < 0.001 (two-tailed). **d** Intersection of gene sets with a significant transcriptomic association with the target pattern across various decoding methods. Gene set intersections and total set sizes are shown at an adjusted *p*-value threshold of 0.05 (left panel), 0.01 (middle panel), and 0.001 (right panel).

transcriptomic maps[3]. In contrast, GLS-decoding resulted in the lowest false positive rate (FPR) and yielded findings that were both sensitive and specific. However, incorporating the full spatial autoregressive correlation structure alongside stringent FDR adjustments may result in overly conservative findings, especially at more conservative levels of statistical stringency, where only a few significant genes were observed (Fig. 3d, Supplementary Data Figs. 2, 3d). Thus, while GLS-decoding seems well suited for hypothesis and enrichment testing, it is less optimal for broader exploratory analyses. Vertex-level decoding is therefore expected to be particularly effective for examining IDPs characterized by highly variable (i.e., high-frequency) signal

fluctuations across the cortical surface, which cannot be fully captured by the native resolution of the AHBA.

Last, we compared the gradient-based approach with precomputed spatial nulls against a model where N = 1000 variogram matching permutations[18] or spins[21] of the target pattern were generated. We observed a strong correspondence between the genes identified as significant by the gradient- and permutation-based approach. More specifically, all genes identified by permuting the target patterns were also identified by the gradient-based approach. In contrast, the spin model was not sufficiently conservative to reliably distinguish the target gene from the gene background, even at more conservative *p*-

value thresholds (see Supplementary Data Figs. 4–6). Given its high sensitivity, specificity, and computational efficiency, we employed the gradient-based approach as the method of choice for all subsequent analyses.

## Two classes of GABA$_A$-receptor subunits with different mRNA expression signatures

Next, we transcriptomically decoded a high-resolution, surface-based in vivo PET atlas of GABA$_A$-receptors (GABA$_A$Rs) targeting benzodiazepine (BZ) binding sites (Fig. 4a)[22]. GABA$_A$Rs are encoded by 13 subunit genes that are reliably expressed in cortical brain tissue (i.e., $\alpha_{1-5}$, $\beta_{1-3}$, $\gamma_{1-3}$, $\delta$, $\varepsilon$; see Methods for details). Across decoding techniques, seven GABA$_A$R subunit genes significantly correlated with BZ binding sites based on their cortical expression signatures (Fig. 4d). More specifically, a significant vertex-level spatial correlation was observed for the $\gamma_3$-subunit gene *GABRG3* ($r = 0.48$, $p_{\text{perm.adj}} < 0.005$), and for the $\beta_2$-subunit gene *GABRB2* ($r = 0.395$, $p_{\text{perm.adj}} < 0.05$). The LME-decoding approach also established a significant (i) positive correlation between mRNA expression and receptor density for the $\gamma_2$- and $\alpha_4$-subunit genes *GABRG2* and *GABRA4*, and (ii) a negative association for the $\beta_4$-subunit gene *GABRB4* and the $\alpha_{2,5}$-subunit genes *GABRA2* and *GABRA5* (Fig. 4d). However, while the mRNA expression patterns of these subunit genes all correlated with BZ-binding sites, there was considerable regional variation within subunit expression signatures across the cortex. (Supplementary Data Fig. 7a). This indicates that different GABA$_A$R subunits spatially align with distinct neural systems and, thus, specific behavioral outcomes. We therefore subsequently stratified GABA$_A$R subunits based on their gene expression profiles.

To delineate the neural systems with high expression levels of specific GABA$_A$R subunits, we employed a hierarchical clustering approach that grouped subunits based on similarities in their vertex-level mRNA expression profiles. The clustering algorithm identified two classes of GABA$_A$R subunit genes, each with a distinct cortical expression pattern (Fig. 4e), with a mean bootstrapped Jaccard similarity index exceeding 0.9 (see Methods for details). The first cluster (Cluster 1) contained GABA$_A$R subunits $\alpha_{2,3,5}$, $\beta_{1,3}$, as well as $\varepsilon$ and $\gamma_1$. This cluster was characterized by (i) elevated mRNA levels primarily in limbic brain regions, including the anterior temporal lobes, entorhinal cortex, anterior insular cortex, and medial orbitofrontal cortex[23], and (ii) low expression in the occipital lobes and the precentral gyrus (Fig. 4e). The second cluster (Cluster 2) included subunits $\alpha_{1,4}$, $\beta_2$, $\gamma_{2,3}$, and $\delta$ (Fig. 4e), and displayed elevated mRNA expression across the cortical hemisphere, particularly in the occipital lobes, except in regions where Cluster 1 subunits were highly expressed. GABA$_A$R subunits can thus be subdivided into two distinct co-expression classes, each defined by a unique mRNA expression signature across the cortical surface.

Notably, while the spatial interpolation increased the magnitude of the correlations between the 13 GABA$_A$Rs subunits overall, it did not alter the relative positioning of genes. This indicates that the differential stability of genes at the vertex-level accurately reflects their stability at AHBA sample resolution (i.e., prior to interpolation) (Supplementary Data Fig. 8).

## Stratification of imaging phenotypes based on their transcriptomic alignment with GABA$_A$R subunit genes

To link genes within GABA$_A$R co-expression clusters to behavioral variation in vivo, we examined the spatial correlations between the IDPs of $N = 279$ males and females, aged 7–31 years, who had available anxiety and depression scores, and the expression patterns of the 13 GABA$_A$R subunit genes. Sample characteristics are provided in the Methods section. To make individuals comparable, IDPs were standardized within the normative (i.e., neurotypical) range to account for the effects of age, sex, full-scale IQ (FSIQ), and other measures affecting brain structure (see Methods for details). Hence, instead of

analyzing absolute CT metrics, all datasets were standardized relative to the canonical trajectory of brain development (Fig. 5a). Using hierarchical clustering, IDPs were then stratified according to their spatial similarity (i.e., neuroanatomical affinity) with GABA$_A$R subunit classes.

Across multiple validity indices (see Methods), we discerned an optimal bifurcated clustering solution with a mean bootstrapped Jaccard similarity index of 0.714 for the primary cluster, and of 0.591 for the secondary cluster. Accordingly, our cohort was divided into two neuroanatomically distinct subgroups, each showing a different neuroanatomical association with the limbic and cortical expression signatures of GABA$_A$R subunit Clusters 1 and *2* (Fig. 5a). Subgroup 1 consisted of 178 individuals (65 females, 113 males) where positive CT deviations (i.e., greater CT than expected) were associated with increased mRNA levels in the limbic GABA$_A$R subunit Cluster 1, and (ii) negative CT deviations (i.e., less CT than predicted) were associated with increased mRNA levels in the cortical GABA$_A$R subunit Cluster 2. Subgroup 2 consisting of 101 individuals (36 females, 65 males) and was characterized by positive correlations with the cortical Cluster 2, and by negative correlations with the limbic GABA$_A$R subunit Cluster 1. IDPs can therefore be distinguished based on their spatial alignment with the cortical expression signatures of two GABA$_A$R subunit clusters: one with a more limbic pattern of expression, and the other with an unspecific, region-overarching pattern of expression.

## Differences in transcriptomic alignment with GABA$_A$R subunit genes are associated with distinct behavioral phenotypes

To link the transcriptomically-derived subgroups to distinct clinical/behavioral phenotypes, we performed a comparative subgroup analysis using measures of anxiety and depression, both of which can be modulated through pharmacological interventions targeting GABAergic and/or serotonergic signaling pathways. Individuals in Subgroup 1 exhibited significantly elevated self-reported levels of anxiety ($t(74) = 2.52$, $p < 0.01$, $p_{\text{adj}} = 0.03$, one-tailed) and depression ($t(73) = 3.65$, $p < 0.001$, $p_{\text{adj}} < 0.01$, one-tailed) compared to those in Subgroup 2 (Fig. 5b). No significant differences in self-reported anxiety or depression levels were observed between subgroups among adolescents, nor in parent-reported measures for children (Fig. 5b). Subgroups also did not differ significantly in terms of age ($t(193) = 0.35$, $p = 0.36$, one-tailed), FSIQ ($t(215) = -0.07$, $p = 0.52$, one-tailed), sex ($\chi^2(1) < 0.001$, $p > 0.99$), or age-groups ($\chi^2(3) = 0.284$, $p = 0.96$).

In adults, we observed a significant positive correlation between levels of anxiety and neuroanatomical diversity within the limbic GABA$_A$R subunit Cluster 1 mask, which contained subunits $\alpha_{2,3,5}$, $\beta_{1-3}$, $\varepsilon$, and $\gamma_1$ ($r = 0.26$, $t(80) = 2.41$, $p < 0.01$, one-tailed). Here, as predicted from the subgroup analyses above (see Fig. 5b), more positive deflections from the typical CT trajectory were associated with elevated self-reported levels of anxiety (Fig. 5c, left panel). This relationship was absent within the mask representing the more unspecific (i.e., region-overarching) cortical expression pattern of GABA$_A$R subunit Cluster 2, which contained subunits $\alpha_{1,4}$, $\beta_2$, $\gamma_{2,3}$, and $\delta$ ($r = 0.005$, $t(80) = 0.04$, $p = 0.48$) (Fig. 5c, right panel; see Supplementary Data Fig. 9 for model's generalization performance). No significant correlations were observed between variation in CT for levels of depression, and for anxiety/depression scores in children and adolescents (all $p$-values > 0.05).

## Discussion

Here, we utilized an analytical framework for the fast (i.e., computationally efficient) genome-wide transcriptomic decoding of high-resolution surface-based IDPs, using spatially-dense, vertex-level representations of the human brain transcriptome provided by the AHBA. This framework was initially validated against PET targets with a well-documented, strong correspondence between cortical mRNA transcript and protein binding (5-HTRs). We then compared our gradient-based approach to alternative techniques operating at the

a | BZ receptor density (Norgaard et al. 2020)

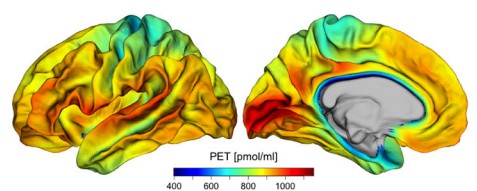

b | Probability of significance

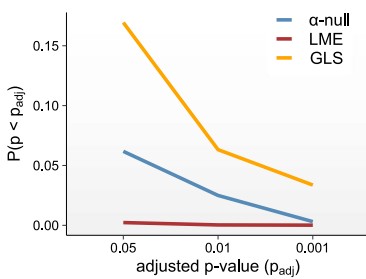

c | Transcriptomic associations of GABA$_A$R subunit genes

| | GABRA3 | GABRA2 | GABRG1 | GABRA5 | GABRB1 | GABRB3 | GABRE | GABRA4 | GABRG2 | GABRA1 | GABRD | GABRB2 | GABRG3 |
|---|---|---|---|---|---|---|---|---|---|---|---|---|---|
| α-null | −0.25 | −0.07 | −0.09 | −0.12 | −0.16 | 0.10 | 0.10 | 0.31 | 0.25 | 0.24 | 0.22 | 0.40* | 0.48* |
| LME | −0.22 | −0.08* | −0.08 | −0.17* | −0.18* | 0.01 | 0.01 | 0.19* | 0.21* | 0.24 | 0.23 | 0.25* | 0.19* |
| GLS | −0.04 | 0.02 | 0.01 | 0.01 | −0.03 | 0.00 | −0.00 | 0.03 | 0.04 | 0.03 | 0.02 | 0.01 | −0.00 |

Subunit: A, B, D, E, G

GABRA2    GABRA5    GABRB1    GABRA4    GABRG2    GABRB2    GABRG3

d | GABA$_A$R subunit clustering

GABA$_A$R subunit Cluster 1

GABA$_A$R subunit Cluster 2

Subunit: A, B, D, E, G
Cluster: 1, 2

GABRA3, GABRA2, GABRA5, GABRB1, GABRE, GABRB3, GABRG1, GABRG3, GABRA4, GABRD, GABRB2, GABRA1, GABRG2

native resolution of the AHBA. Subsequently, we used the vertex-level mRNA expression signatures to dissect two discrete classes of GABA$_A$R subunits, which were spatially aligned with IDPs to probe their specific functional role. Our findings indicate that the cortical transcriptomic landscape of genes encoding for specific pharmacological targets may be indicative of their clinical or behavioral relevance, and so guide the development of targeted pharmacotherapies in the future.

Our study builds on previous work by Gryglewski et al. (2018), who were the first to provide mRNA expression signatures for all protein-coding genes across the entire cortical surface, at a resolution that is comparable to conventional surface-based IDPs[6]. This is absolutely vital as it enables the seamless integration of in vivo imaging data with ex vivo gene expression patterns. Unlike Gryglewski et al. (2018), we performed spatial interpolation based on AHBA samples identified

**Fig. 4 | Surface-based transcriptomic decoding of the Benzodiazepine (BZ) binding sites. a** A high-resolution in vivo atlas of the human brain's BZ binding site of GABA$_A$Rs provided by ref. 22. **b** False positive rate (FPR) of the α-null (blue), Linear Mixed Effects (LME, red), and General Least Squares (GLS, yellow) decoding techniques across different adjusted $p$-value thresholds (p$_{adj}$). For LME- and GLS-decoding, $p$-values were adjusted for multiple comparisons using False Discovery Rate (FDR) adjustments. For α-null decoding, adjusted permutation $p$-values (p$_{perm.adj}$) were used. **c** Upper panel: Transcriptomic associations between BZ binding sites and mRNA expression signatures of individual GABA$_A$R subunits for different decoding approaches. α-null: spatial autocorrelation preserving null model; LME: Linear Mixed Effects model; GLS: General Least Squares model. Lower panel: predicted mRNA expression profile of significant subunits across the cortical surface. * adjusted $p$-value < 0.05 (two-tailed). **d** Hierarchical clustering of GABA$_A$R subunits based on their vertex-level transcriptomic associations. Subunits were allocated to two classes with a distinct pattern of expression. The first cluster (Cluster 1) contained subunit genes *GABRA3*, *GABRA2*, *GABRA5*, *GABRB1*, *GABRE*, *GABRB3*, and *GABRG1* with high predicted expression levels in the limbic circuitry. The second cluster (Cluster 2) contained subunit genes *GABRG3*, *GABRA4*, *GABRD*, *GABRB2*, *GABRA1*, and *GABRG2* with high predicted expression levels across the cortical surface. The left panel shows the predicted mean expression signatures across genes within each subunit cluster.

within a geodesic−rather than Euclidean−vertex neighborhood (see[24,25] for geodesic distance mapping). While the resulting surface-based gene expression signatures are expected to be theoretically more accurate, as they account for the highly complex pattern of cortical folding, they were also highly correlated with the mRNA expression profiles published by Gryglewski et al. (2018). Additionally, the maps showed strong agreement with the spatially-smoothed dense expression patterns recently released by Wagstyl et al. (2023) on a lower-resolution (30 k) surface template[20]. The convergence of patterns across different analytical strategies thus underscores the robustness of these maps and highlights they potential for linking IDPs to underlying mechanisms.

However, linking spatially-dense gene expression patterns to highly variable IDPs, both across individuals and brain regions, is a computational and statistical challenge. Our study shows that spatial interpolation significantly enhanced the spatial correlations between gene expression signatures overall. However, the relative positioning of genes with respect to each other, or their differential stability, remained unchanged. The observed increase in spatial correlations is likely driven by the increase in spatial autocorrelation (i.e., smoothness) within the interpolated patterns. This can lead to an inflation of false-positives when establishing transcriptomic associations between IDPs and genome-wide expression signatures[3]. This was addressed by placing all genome-wise analysis within the spatial null modeling framework[26], where spatial null models are generated to derive an empirical distribution of spatial correlations under the null hypothesis. To overcome the computational challenges associated with the generation of reliable spatial nulls, we reduced the gene expression signatures of all protein-coding genes to a smaller subset of gene co-expression gradients, for which spatial null models were subsequently precomputed. These co-expression gradients closely mirror the transcriptional developmental programs that shape human brain organization[27] and that have recently been described in detail by Wagstyl et al. (2023)[20]. Notably, these large-scale canonical expression patterns of modules not only align with the diverse spatial scales and temporal epochs of human brain organization−ranging from cytoarchitectonic boundaries to markers of neuronal subtypes−but also seems to be functionally relevant[20]. Given the biological plausibility of the model and the large degree of spatial covariation in gene expression patterns, a gradient-based approach represents a valuable alternative to the genome-wide analyses of transcriptomic associations.

In particular, using 5-HTR PET maps[15] as target patterns, we demonstrated that that the sensitivity and specificity of the gradient-based approach compared well to (i) alternative techniques relying on existing AHBA samples exclusively (e.g., LME or GLM decoding that do not employ spatial interpolation), or (ii) approaches where the target pattern is spatially permuted[18] or rotated (see spin model[21]). Moreover, the gradient-based approach is expected to be more reliable than traditional region-based decoding techniques when dealing with IDPs that are characterized by highly variable signal fluctuations across the cortical surface, which cannot fully be captured by the native resolution of the AHBA. This is particularly relevant when examining IDPs in

neuropsychiatric and neurodevelopmental conditions that are marked by highly diverse and individualized neuroanatomical and functional variations in the brain (e.g.[5],). In the current study, we focused on neuroanatomical variations in cortical brain tissue exclusively given their particular importance to neuropsychiatric disorders. Moreover, in vertex-level decoding, spatial correlations are computed across thousands of vertices (>41 k), whereas the number of subcortical brain regions is considerably smaller (∼6−10 per hemisphere). Consequently, the relative the influence of subcortical brain regions on spatial correlations across the brain would be minimal overall.

Next, we examined the cortical mRNA expression signature of the GABAergic system based on 13 subunit genes that encode GABA$_A$R isoforms. Notably, it has previously been shown that GABA$_A$R isoforms exhibit a highly region-specific expression pattern and are differentially correlated with BZ binding sites[12,22]. This allowed us to test the hypothesis that the specific regional expression pattern of a molecular target may be indicative of its functional correlates. Here, we demonstrated that GABA$_A$R subunits genes can be grouped into two distinct classes based on their cortical expression signatures. The first class (Cluster 1) represented the transcriptomic signature of subunits $\alpha_{2,3,5}$, $\beta_{1-3}$, $\varepsilon$, and $\gamma_1$ and was confined to brain regions within the limbic circuitry. The second class (Cluster 2) encompassed genes encoding for subunits $\alpha_{1,4}$, $\beta_2$, $\gamma_{2,3}$, and $\delta$ and exhibited a more general, unspecific expression signature across the cortical surface. These findings agree with previous reports noting a strong co-distribution of subunits $\alpha_1$, $\beta_2$, and $\gamma_2$ across the brain, which was also consistent across donors[12]. Moreover, different GABA$_A$R isoforms show a region-specific pattern of expression, and so may be linked to different behavioral domains[12]. For instance, the high expression of Cluster 1 in the broader limbic neurocircuitry, often referred to as the emotional brain[23], indicates that genes within this cluster are involved in processes subserving socio-emotional functioning. In contrast, the expression of genes in Cluster 2 was not confined to any specific set of brain regions or neural system, indicating that heteromeric GABA$_A$R isoforms containing $\alpha_1\beta_2\gamma_2$ subunits may be essential for mediating a broader spectrum of neurocognitive functions.

To probe the putative functional role of GABA$_A$R subunit classes, we aligned the neuroanatomical IDPs of 279 individuals to their cortical expression gradients by means of spatial correlation. Using hierarchical clustering, we established that IDPs can be subdivided based on their differential neuroanatomical association with GABA$_A$R subunit expression patterns. Individuals in Subgroup 1 displayed a pattern of CT variability that positively correlated with the limbic GABA$_A$R subunit Cluster 1. IDPs of individuals in Subgroup 2 were positively correlated with the co-expression signatures of cortically-expressed GABA$_A$R subunit genes in Cluster 2. As the developmental trajectory of CT has an inverted U-shape across the human life span[28], positive deviations from the typical trajectory of CT are commensurate with delayed brain maturation. In line with this, individuals − and adults in particular − with more atypical CT in the limbic brain circuitry, which was characterized by high expression of the $\alpha_2$-containing GABA$_A$R subunit Cluster 1, also had significantly higher levels of anxiety and depression than adults falling into the $\alpha_1$-containing GABA$_A$R subunit

**a | Mapping imaging phenotypes to GABA$_A$ receptor subunits**

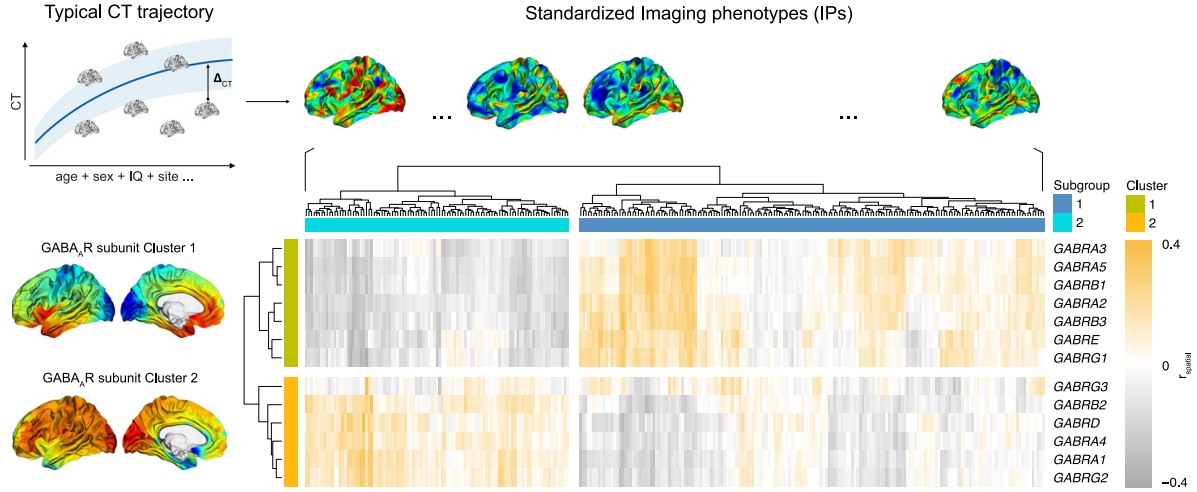

**b | Subgroup differences in anxiety and depression**

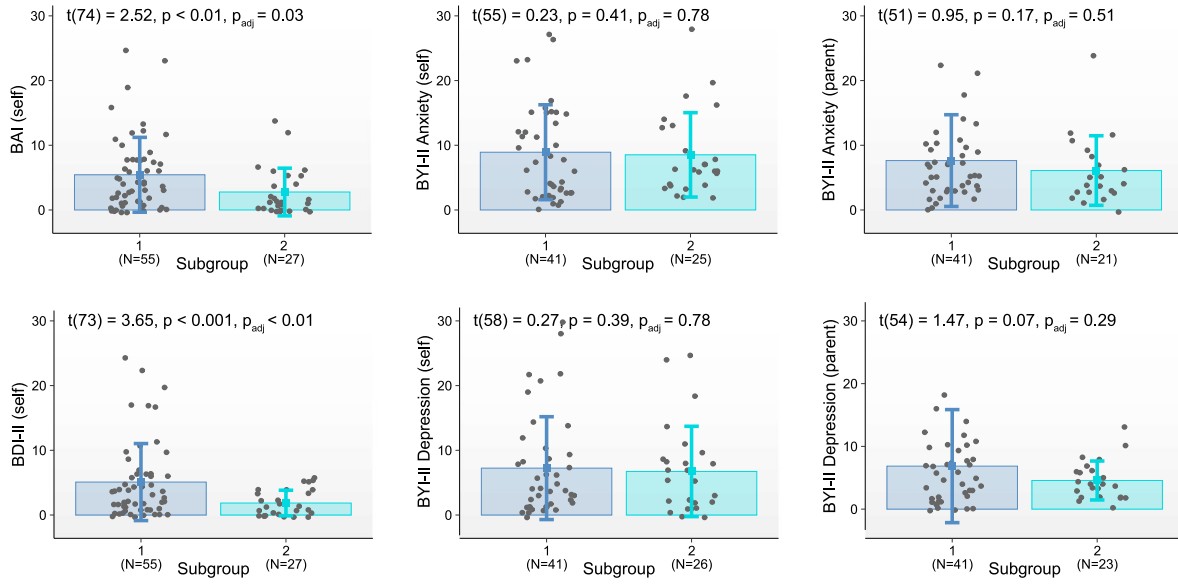

**c | Brain-behavioral correlations within GABA$_A$R subunits systems**

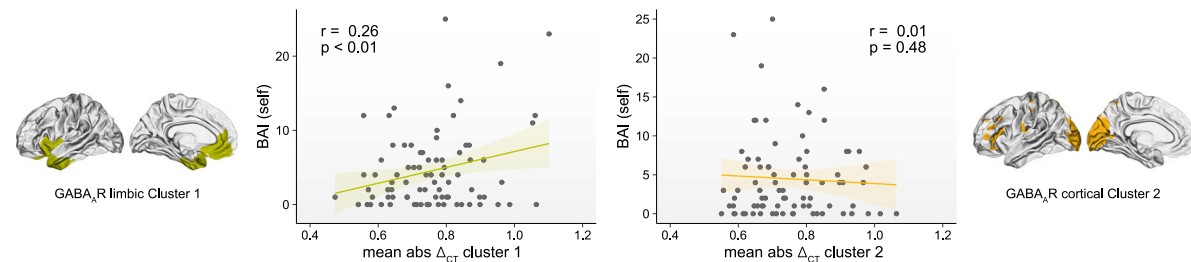

Cluster 2. Several factors could explain why these correlations were observed in adults but not in children and adolescents. One possibility is a discrepancy between self-reported and parent-reported levels of depression and anxiety, which appears to diminish with age[29]. Additionally, both the AHBA and PET atlas data are derived from adult samples. Thus, transcriptomic associations may be more accurate in adult populations compared to younger age groups.

Our finding also aligns with previous evidence suggesting that BZ binding sites can be categorized based on $\alpha$ subunit isoforms and their specific behavioral implications (reviewed in ref. 13). In brief, the $\alpha_1$ subunit is associated with BZ type I (BZ1) pharmacology and has been shown to have primarily sedative, amnesic, and anticonvulsive effects. The $\alpha_2$ subunit, on the other hand, is associated with BZ type II (BZ2) pharmacology and mediates the anxiolytic actions of

**Fig. 5 | Stratification of imaging phenotypes based on their transcriptomic association with GABA_A-receptor subunits. a** Upper panel: Standardization of neuroanatomical imaging-derived phenotypes (IDPs) based on the neurotypical trajectory of cortical thickness (CT). All IPs were transformed to unit standard deviations relative to the canonical trajectory of CT ($\Delta_{CT}$). Data are presented as mean ± standard deviation of residuals across individuals. Note. For illustration purposes only. Lower panel: hierarchical clustering of IPs based on their transcriptomic association with GABA_AR subunits. **b** Differences in levels of anxiety and depression between subgroups with a differential transcriptomic association with GABA_AR subunits. BAI: Beck Anxiety Inventory; BYI-II: Beck Youth Inventories 2nd

Edition; BDI-II: Beck Depression Inventory 2nd Edition; self: self-reports; parents: parent-reports. Error bars present mean values ± standard deviations. **c** Brain-behavioral correlations between self-reported measures of anxiety in adults and the individuals' total degree of neuroanatomical diversity in CT within brain regions with high mRNA expression levels of GABA_AR subunit Cluster 1 and Cluster 2. Masks were generated by applying a threshold to the mean expression patterns of genes within each cluster, anchored at the 80th percentile of their distribution across the cortex. Shaded area indicates the 95% confidence interval around the regression line.

benzodiazepines. Our study of cortical gene expression patterns extends this previous evidence by proposing that the distinct behavioral effects of $\alpha_1$ and $\alpha_2$ may be mediated through two different neural systems. This means that the cortical expression signatures of $\alpha_1$ and $\alpha_2$ subunit genes are largely non-overlapping (negatively correlated), with $\alpha_2$ predominantly affecting limbic system functions, and $\alpha_1$ impacting on more general, region-overarching neural motives. It is important to note, however, that the magnitude of the spatial correlations in gene expression primarily depends on the degree of spatial coherence in mRNA levels across the cortical surface (i.e., coherence in regional variations), rather than similarities in absolute mRNA expression levels. A gene with persistently high expression levels across the cortical surface may therefore exhibit only a weak correlation with a regionally highly variable IDP, yet still have a significant impact on a phenotype. Thus, the impact of a gene on a phenotype cannot be inferred solely based on spatial correlation.

Along the same lines, although CT variability has previously been related to genetic variation in GABA-ergic genes[30], this does not necessarily imply a causal relationship between GABA-ergic regulation and the development of this particular aspect of the neural architecture (i.e., thickness of the cortical mantle). In the future, it will therefore be important to explore the causal mechanisms and pathways that link genes to IDPs more closely, also based on other morphometric features that are under close genetic and/or transcriptional control[31]. Additionally, it is important to recognize that the associations between IDPs and target patterns, as well as their relationships with behavioral profiles, are currently derived from in-sample estimates from a cross-sectional cohort, rather than out-of-sample predictions. The crucial next step will be to apply our model to longitudinal datasets, both before and after pharmacotherapy, to assess its potential for guiding personalized treatment and intervention strategies.

## Methods

### Ethical approvals
The study was approved by national and local ethics review boards at each site, and was carried out to Good Clinical Practice (ICH GCP) standards. More specifically, at each recruitment center, namely (i) King's College London & University of Cambridge, London-Central and Queen Square Health Research Authority, Research Ethics Committee (Ref. Nr. 13/LO/1156), (ii) Radboud University Medical Centre & University Medical Centre Utrecht, Institute Ensuring Quality and Safety Committee on Research Involving Human Subjects Arnhem-Nijmegen (Ref. Nr. 2019-5942), (iii) Medical University Mannheim, Medical Ethics Commission II (Ref. Nr. 2020-547 N), and (iv) Bio-Medical University Campus Rome, Ethics Committee De Roma (Ref. Nr. 18/14 PAR ComET CBM). Written informed consent was obtained for all participants. Participants received compensation for study visits.

### Statistics and reproducibility
**AHBA microarray data normalization and FreeSurfer fsaverage6 surface alignment.** Regional microarray expression data was obtained from the six post-mortem brains (1 female, aged 24−57 years) provided by the Allen Human Brain Atlas (AHBA, https://

human.brain-map.org[2]). The expression data was initially pre-processed with the abagen toolbox (version 0.1.1; https://github.com/rmarkello/abagen) using the FreeSurfer fsaverage6 (41k) as standard-space MRI template. The toolbox has been described in detail elsewhere (see refs. 1,32). In brief, probes were initially re-annotated using gene annotations provided by ref. 1. As we restricted our analyses to cortically expressed genes, all tissue samples not assigned to the fsaverage6 cortical mask label were discarded. Inter-donor variation was addressed by normalizing expression values across genes using a robust sigmoid function[33]:

$$x_{norm} = \frac{1}{1 + \exp\left(-\frac{x - \langle x \rangle}{IQR_x}\right)} \tag{1}$$

where $\langle x \rangle$ is the median and $IQR_x$ is the normalized interquartile range of the expression of a single tissue class (e.g., cortex) across genes. Normalized expression values were then rescaled to unit interval using

$$x_{scaled} = \frac{x_{norm} - \min(x_{norm})}{\max(x_{norm}) - \min(x_{norm})} \tag{2}$$

Gene expression values were then normalized across tissue samples using an identical procedure. This resulted in a normalized mRNA expression matrix of $N = 1670$ samples (i.e., spatial locations or well IDs) and $N = 15,633$ genes across donors.

The normalized AHBA data was then mapped onto the fsaverage6 surface using an optimized mapping approach via FreeSurfer surface reconstructions of each of the six donor brains (downloaded from https://www.repository.cam.ac.uk/handle/1810/265272). Details of this approach are provided elsewhere (see https://surfer.nmr.mgh.harvard.edu/fswiki/CoordinateSystems). The medial wall label (i.e., subcortical regions and ventricles) was masked out, so that samples were only assigned to cortical vertices. Given the asymmetrical distribution of AHBA samples between hemispheres (i.e., only 2 out of 6 donors have right-hemisphere data), samples were mirrored across hemispheres to maximize the number of data points per hemisphere for spatial interpolation. FreeSurfer xhemi tools were used to map each vertex from one hemisphere to the other via the spherical symmetric fsaverage6 hemispheric surface template (see fsaverage_sym[34]). mRNA expression values of samples assigned to identical vertices were averaged across samples and donors.

**Spatial interpolation of mRNA expression levels across the cortical surface.** For the transcriptomic decoding of surface-based IDPs, we initially generated a spatially-dense representation of the AHBA human brain transcriptome. To generate smooth vertex-level mRNA expression patterns, we adopted a similar approach as described by Gryglewski et al. (2018)[6], whereby mRNA expression values for vertices without AHBA representation are predicted through spatial interpolation using Gaussian Process Regression (i.e., ordinary kriging[35]). To account for the highly complex pattern of cortical folding, we performed spatial interpolation using existing AHBA samples located within a 40 millimeters geodesic distance from each vertex on the FreeSurfer fsaverage6 (41k) surface template (Fig. 1, also see ref. 24).

This distance was chosen to obtain a sufficient number of data points for the prediction of data at unsampled vertices (i.e., $N_{samples} > 30$) while maintaining computational efficiency. At each cerebral vertex not represented in the AHBA, mRNA expression values were then predicted using the 3D coordinates and mRNA data of existing samples within the patch using R for statistical computing (www.r-project.org, version 4.1.2). To this end, we used the autofitVariogram function implemented in the R automap package (version 1.0.14[36]) with a Gaussian kernel and a minimum number of 10 data points per distance bin to automatically fit a variogram based on the existing data. The best-fitting model was subsequently used for the prediction of the estimated mRNA intensity at that vertex. To evaluate the robustness of the predicted mRNA profiles, spatial interpolation was conducted separately for the left and right hemispheres. This resulted in two almost identical matrices, each containing $N = 40,962$ predicted (i.e., spatially interpolated) mRNA expression values at every vertex on the left and right hemisphere, for each of $N = 15,633$ genes (see Supplementary Data Fig. 10). The resulting patterns were smoothed within a 5 mm vertex neighborhood to reduce the effect of outliers.

Our approach utilizing geodesic distances thus converged from that of Gryglewski et al. (2018), who conducted spatial interpolation using the nearest AHBA samples in Euclidean space. Despite these methodological differences, however, the resulting mRNA expression profiles were highly consistent across approaches (see Supplementary Data Fig. 11) and also showed strong agreement with the spatially smoothed dense expression patterns reported by Wagstyl et al. (2023) at a 30k vertex resolution[20] (Supplementary Data Fig. 12).

**Transcriptomic decoding of surface-based IDPs using spatial autocorrelation-preserving null models.** Given the sensitivity and technique-dependent nature of examining spatial associations between gene expression patterns and IDPs[37], we implemented and compared various analytical approaches for the statistical assessment of spatial correlations between high-resolution imaging patterns and genome-wide expression profiles (i.e., gene expression decoding). For the vertex-level decoding using whole-brain mRNA expression patterns, we placed hypothesis testing within the general framework of $\alpha$-preserving null modeling[26], which is designed to account for the large degree of spatial autocorrelation ($\alpha$) inherent to spatially-embedded signals. Here, values in the target pattern are either randomly permuted and subsequently smoothed to reintroduce $\alpha$ characteristics of the original non-permuted data (see variogram matching surrogates[18]), or randomly rotated across the cortical surface (see spin model[21]). Given the high computational demands of the variogram matching approach, it was not easily possible to generate surrogate mRNA expression patterns for each of 15,633 abagen genes. We therefore employed Principal Component Analysis (PCA) to decompose the normalized mRNA signatures of 15,633 genes into a smaller subset of nine co-expression gradients with an Eigenvalue larger than one, capturing ~41% of the total variability in gene expression across the cortex (see Supplementary Data Fig. 13a, b). Genes were subsequently assigned to gradients according to their highest absolute loadings, i.e., spatial correlation between each gene and gradient patterns.

For each gradient pattern, a total of $N = 1000$ $\alpha$-preserving null models were then precomputed according to[18] to characterize the empirical distribution of spatial correlations under the null hypothesis (Fig. 2b). More specifically, gradient null models were computed using the BrainSMASH toolbox (https://github.com/murraylab/brainsmash) with resampling under the exclusion of vertices in the medial wall label. Notably, for each gradient, we optimized the $k_{nn}$ parameter by minimizing the Normalized Root Mean Squared Error (NRMSE) between the fitted variograms of 100 surrogate maps and the target map's variogram across $k_{nn}$ $\varepsilon$ {1000; 5000; 1000; 20,000; 30,000; 40,000}. This optimization step is vital, as the $k_{nn}$ parameter

determines the number of nearest neighbors (i.e., data points) for variogram fitting, and so directly impacts on the quality of the spatial prediction. We identified an optimal $k_{nn}$ of 40,000 for PC$_{1-2}$, of 30,000 for PC$_3$, of 20,000 for PC$_{5-7}$, and of 10,000 for PC$_{4,8-9}$ (Supplementary Data Fig. 13c). To validate the use of gradient-based nulls as proxies for individual gene expression patterns, we compared empirical variograms of the nine co-expression gradients with those of individual gene's expression signatures. The analysis across a random subset of genes revealed a strong alignment between variograms, indicating that the spatial autocorrelation inherent in gradient patterns and their surrogates closely mirrors the spatial dependence observed in individual gene expression signatures (see Supplementary Data Fig. 14). These findings support the use of co-expression gradients as surrogates for individual gene expression signatures to derive an empirical distribution of spatial correlations under the null hypothesis.

A non-parametric $\alpha$-corrected $p$-value estimate ($p_{perm}$) for each gene was therefore derived based on the null distribution of spatial correlations with its target map across the pre-computed nulls for the respective gradient pattern (Fig. 2c). Gene-level $p$-values were adjusted for multiple comparisons ($p_{perm.adj}$) using the empirical cumulative density function of the extreme value distribution of spatial correlations across genes within gradient permutations (Fig. 2c). This approach is comparable to the maxT algorithm proposed by Westfall & Young (1993)[38] and the fast permutation inference approach published by ref. 39. However, instead of selecting the maximal test statistic across feature permutations, we modeled the empirical extreme value distribution across gradient patterns.

To evaluate the statistical rigor of the model with precomputed gradient surrogates, we also generated $N = 1000$ spatial null models for the target pattern (i.e., mRNA expression signatures of 5-HT$_{1A}$R, 5-HT$_{2A}$R, and 5-HT$_4$R) according to (i) Burt et al. (2020) with optimization of the $k_{nn}$ parameter[18], and (ii) employing the spatial permutation null or spin modeling approach proposed by Alexander-Bloch et al. (2018)[21]. This allowed us to adjust permutation $p$-values based on the empirical cumulative density function of the extreme value distribution of spatial correlations across genes, rather than gradient patterns. Moreover, to estimate the impact of the spatial interpolation on the magnitude of spatial correlations between genes and the relative differential stability of genes, we compared the matrix of spatial correlations of the mRNA expression patterns for the 13 GABA$_A$R subunit genes examined in the present study before and after spatial interpolation.

**Transcriptomic decoding using Linear Mixed Effects (LME) and General Least Squares (GLS) modeling.** To assess the effects of the spatial interpolation and the robustness of our findings, we compared the gradient-based approach to alternative techniques that were based solely on existing AHBA data (i.e., without spatial interpolation). This included a Linear Mixed Effects (LME) model and General Least Squares (GLS) decoding. Here, the abagen preprocessed AHBA data was initially mapped onto the FreeSurfer fsaverage6 as described above. To align the MRI and gene expression data, the target pattern was downsampled to match the native resolution of the AHBA by averaging vertex values within a 5 mm geodesic neighborhood around each of $N = 1670$ AHBA sampling sites (Supplementary Data Fig. 1a). To identify genes significantly correlated with the target pattern, a Linear Mixed Effects (LME) model with random intercept and slope grouped by donor (LME-decoding) was fitted for each gene using the lme4 package in R (version 1.1.28). This approach was originally proposed by ref. 19 and implemented within the Neurosynth platform (https://neurosynth.org), and highlights genes that are consistently highly correlated with the target pattern across donors (Supplementary Data Fig. 1b). Uncorrected gene $p$-values associated with the fixed effect of mRNA expression were identified using the Satterthwaite

approximation for degrees of freedom as implemented in the R package afex (version 1.0.1).

The LME model does not account for spatial autocorrelations, as statistical effects are based on consistencies across donors. As a result, fewer (i.e., within-donor) samples that are further apart are considered when quantifying the relationship between gene expression and target patterns. We therefore also implemented a Generalized Least Squares (GLS) decoding approach that accounted for the full spatial auto-correlation structure. This model predicted the target map by the mRNA profile of each gene, covarying for a Gaussian autoregressive spatial correlation structure defined by vertex x,y,z coordinates, and donor as grouping factor (Supplementary Data Fig. 1c). The GLS model was fitted using the R package nlme (version 3.1.153). Gene-level $p$-values were obtained based on the main effect of mRNA expression. For both LME- and GLS-decoding, gene-level $p$-values were corrected using a False-Discovery Rate (FDR) adjustment ($p_{adj}$).

**Comparison of decoding techniques using in vivo PET atlas data.** To assess the validity of the predicted vertex-level mRNA expression maps, and to examine the sensitivity and specificity of the various models, we utilized publicly available Positron Emission Tomography (PET) atlas data of molecular targets that (1) are highly expressed in cortical brain tissue, and (2) display a significant high correlation between mRNA expression and protein density (e.g refs. 14,40.). This includes high-resolution in vivo PET atlas data of the human ser-otonergic (5-HT) system, which has been released on the FreeSurfer fsaverage surface for four 5-HT receptors (i.e., 5-HT$_{1A}$, 5-HT$_{1B}$, 5-HT$_{2A}$, 5-HT$_4$), and for the 5-HT transporter (5-HTT)[15]. We did not examine the 5-HT transporter (5-HTT), which is predominantly expressed in sub-cortical brain regions. The 5-HT$_{1B}$R gene *(HTR1B)* did not survive aba-gen quality assessments and was also excluded from the analyses. Moreover, we utilized an in vivo PET atlas of the human brain's ben-zodiazepine (BZ) binding sites located on postsynaptic GABA$_A$Rs[22]. GABA$_A$Rs have a highly complex pentameric structure formed by 19 different genes that encode eight distinct subunit classes (reviewed in ref. 41). Of the 19 subunit genes encoding GABA$_A$-Rs (i.e., $\alpha_{1-6}$, $\beta_{1-3}$, $\gamma_{1-3}$, $\rho_{1-3}$, $\delta$, $\varepsilon$, $\vartheta$, $\pi$; see Fig. 5b), we were able to reliably quantify and analyze the cortical mRNA expression signatures of 13 subunits (i.e., $\alpha_{1-5}$, $\beta_{1-3}$, $\gamma_{1-3}$, $\delta$, $\varepsilon$) via the AHBA. The remaining six subunits are mostly expressed in non-cortical brain tissues (see Human Protein Atlas (HPA) for information on regional expression patterns, www.proteinatlas.org). Data was downloaded from https://xtra.nru.dk/BZR-atlas/ and https://xtra.nru.dk/FS5ht-atlas/ respectively.

For these molecular targets, we initially examined the quality of the predicted vertex-level mRNA expression maps by assessing their spatial correlation ($r_{spatial}$) with the target's non-displaceable binding potential (BPND). The significance of each spatial correlation was established as described above. The sensitivity and specificity of the models were then evaluated based on a model's ability to detect a significant correlation if it exists (i.e., $p(r_{spatial}) < 0.05$), and the model's ability to detect a significant correlation in the context of all other genes (i.e., $p_{adj}(r_{spatial}) < 0.05$), respectively. Moreover, for each model, we examined the number of false positives across different levels of statistical stringency, based on the probability of obtaining significant genes, i.e., $P(p < p_{adj})$.

**Hierarchical clustering of GABA$_A$R subunits genes based on their cortical expression patterns.** A hierarchical clustering approach was employed to cluster 13 cortically expressed GABA$_A$R subunits (i.e., $\alpha_{1-5}$, $\beta_{1-3}$, $\gamma_{1-3}$, $\delta$, $\varepsilon$) based on the normalized predicted mRNA expression profile of corresponding subunit genes (i.e., *GABRA1* to *GABRA5*, *GABRB1* to *GABRB3*, *GABRD*, *GABRE*, and *GABRG1* to *GABRG3*) across the cortical hemisphere. A distance matrix $D(i,j) = 1 - r_{i,j}$ was then computed to serve as input for hierarchical clustering, where $r_{i,j}$ denotes the Pearson correlation between the expression patterns of

subunits $i$ and $j$. The spatial correlation matrix of GABA$_A$R subunit genes is shown in Supplementary Data Fig. 7b. The optimal number of clusters was identified using the R package NbClust (version 3.0.1) through the complete aggregation method, which evaluates clustering solutions for different numbers of clusters across multiple validity indices (frey, mcclain, cindex, silhouette, and dunn index). The stability of the clustering solution was evaluated using the clusterboot function implemented in the R package fpc (version 2.2.9). This function performs bootstrap resampling of the data and evaluates the stability of each cluster by calculating Jaccard coefficients, which represent the similarity between the original clustering and the boot-strapped clustering. Note that a Jaccard similarity value of 0.75 or more generally indicates a stable clustering solution.

**Transcriptomic alignment between IDPs and GABA$_A$R gene expression patterns.** To probe the putative functional involvement of GABA$_A$R subunits, we spatially aligned the cortical mRNA expression signatures of the respective genes with the neuroanatomical IDPs of 279 individuals. These data were derived from participants enrolled in the EU-AIMS Longitudinal European Autism Project (LEAP; www.aims-2-trials.eu)[42]. They included 254 typically developing participants (90 female, 164 male) and 25 individuals with mild intellectual disability (ID; 11 female, 14 male; defined by a Full-Scale IQ (FSIQ) between 50 and 74) between the ages of 7 and 31 years (mean age = 17.32 ± 5.91 years). Note that we relied on self-reported measures of biological sex, rather than gender. The original study's sample size was derived through power calculations tailored to that study's aims and objectives[42]; here, we restricted our sample to those participants with high-quality structural magnetic resonance imaging and clinical data. Hence, no statistical method was used to predetermine sample size. The experiments were not randomized and the investigators were not blinded to allocation during experiments and outcome assessment.

For children younger than 11 years, parents completed the depression and anxiety subscales of the Beck Youth Inventories (BYI-II[43]). Adolescents (aged 12–17 years) were given the depression and anxiety subscales of the BYI-II as self-report. In adults, self-reports of symptoms associated with depression and anxiety were measured using the Beck Depression Inventory−Second Edition (BDI-II[44]), and the Beck Anxiety Inventory (BAI[45]), respectively. Further details on summary scores, medication status, and co-occurring mental health conditions (e.g., ref. 46) are provided in Supplementary Tables 1–3. The T$_1$-weighted structural Magnetic Resonance Imaging (MRI) data (see ref. 5 for acquisition parameters) was initially preprocessed using the default pipeline implemented in the FreeSurfer v6.0.0 software (http://surfer.nmr.mgh.harvard.edu/), and quality assessed as outlined in ref. 5. We examined measures of cortical thickness (CT), which represent the closest distance from the outer (i.e., pial) to the inner (i.e., white) matter boundary at each vertex on the tessellated surface[47], smoothed using a 15-mm kernel.

To make individuals comparable, IDPs were initially standardized within the neurotypical (i.e., non-ID) range by means of a General Linear Model (GLM) that included age, sex, FSIQ, acquisition site, and total brain volume as predictors. The model coefficients were subsequently used to predict CT across the cortex for all individuals in our sample, and the resulting residuals were centered and scaled. Thus, instead of employing absolute CT metrics, all datasets were normal-ized to unit standard deviations relative to the canonical develop-mental trajectory. Here, positive values indicated increased CT relative to the expected neurotypical range, while negative values indicated decreased CT. This approach was motivated by so-called normative modeling frameworks[48], which place each individual within a norma-tive range of expected neurotypical variation. These studies show that a person's neuroanatomy is marked by highly individualized patterns of neuroanatomical variability, which may serve as a distinct neuroa-natomical fingerprint that may be utilized for stratification purposes

(e.g. refs. 49, 50). The standardized IDPs were subsequently mapped onto the mRNA expression patterns of GABA$_A$R subunit genes via vertex-level spatial correlations; i.e., Pearson correlation coefficients across the entire cortical surface. To stratify individuals based on their spatial alignment with GABA$_A$-receptor subunit genes, the matrix of spatial correlations was then subjected to hierarchical clustering as outlined above, i.e., using NbClust to identify the optimal number of clusters, and clusterboot to establish their stability. Notably, this approach diverged from using a correlation matrix as input to the clustering algorithm but instead identified consistent patterns of high/low spatial correlations across individuals.

Utilizing the composite expression signatures of the GABA$_A$R subunit Cluster 1 and 2, we also generated a binary mask to demarcate cortical regions with high levels of mRNA expression for cluster (Fig. 5c), allowing us to test the hypothesis that neuroanatomical variability within these regions is associated with differences in behavioral phenotypes. These masks were constructed by applying a threshold to the mean mRNA expression patterns of genes within each cluster, anchored at the 90th percentile of their distribution across the cortical surface. For each participant, we subsequently identified the mean degree of neuroanatomical diversity, quantified as the average difference from the neurotypical CT trajectory across vertices within each mask. These network-specific composite measures of neuroanatomical variation were subsequently correlated with anxiety and depression levels using a linear model. To assess the model's generalization performance (i.e., predictive value), we also placed model fitting within a machine learning (ML) framework. Here, the dataset was randomly split into a training set (75%) and a test set (25%). A linear regression model was initially fitted based on the training set, with the intercept optimized via 3-fold cross-validation. The model's predictive performance was then assessed on the test set. BAI scores were predicted using two models: (i) based on the matrix of spatial correlations between IDPs and the mRNA expression signatures of the 13 GABA$_A$R subunit genes, and (ii) based on the individuals' total degree of neuroanatomical diversity in CT within brain regions with high mRNA expression levels of GABA$_A$R subunit Cluster 1 (see Supplementary Data Fig. 9).

### Reporting summary
Further information on research design is available in the Nature Portfolio Reporting Summary linked to this article.

## Data availability
The raw clinical and neuroimaging data from the LEAP study are protected under data privacy regulations and cannot be publicly shared. The complete dataset generated by the AIMS-2-Trials consortium has been deposited in the Institut Pasteur's secure data repository, Owey (https://dataset.owey.io/doi). Access to the data can be requested on reasonable grounds; for details, please refer to the contact information provided at https://www.aims-2-trials.eu/. Anonymized, pre-processed standardized imaging phenotypes, associated phenotypic data, and spatially interpolated mRNA expression patterns presented and analyzed in this study are provided in a publicly accessible repository (see https://gin.g-node.org/sphache/DecodeGABAData).

## Code availability
All code used for the analyses presented in this study is publicly available on github. The 'fsnulls' package, which implements spatially-constrained null models for surface-based brain imaging data, is available at https://github.com/christineecker/fsnulls. The 'fsdecode' package, which supports gene expression decoding of imaging phenotypes using data from the Allen Human Brain Atlas, is available at https://github.com/christineecker/fsdecode. Both packages are openly licensed and include documentation and example scripts to facilitate reproducibility. All scripts for clustering IDPs based on their

neuroanatomica affinity to GABA-ergic genes can be assessed at https://gin.g-node.org/sphache/DecodeGABA/src/main.

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

## Acknowledgements

We gratefully acknowledge the contributions of the EU-AIMS LEAP Group (see Supplementary Information for full list of consortium members). This project has received funding from the Innovative Medicines Initiative 2 Joint Undertaking under grant agreement No 115300 for the project EU-AUIMS and No 777394 for the project AIMS-2- TRIALS. This Joint Undertaking receives support from the European Union's Horizon 2020 research and innovation program and EFPIA and AUTISM SPEAKS, Autistica, SFARI. Any views expressed are those of the author(s) and not necessarily those of the funders (IHI-JU2). This work has also received funding from Horizon Europe (Grant Agreement No. 101057385) and from UK Research and Innovation (UKRI) under the UK government's Horizon Europe funding guarantee (Grant No. 10039383) (R2D2-MH). CE gratefully acknowledges support from the German Research Foundation (DFG) (Project Nr. 551579379), the Rolf M. Schwiete Stiftung (Project Nr. 2024-022), and the DYNAMIC center funded by the LOEWE program of the Hessian Ministry of Science and Arts (Grant Number: LOEWE1/16/519/03/09.001(0009)/98). DGM also acknowledges support from the NIHR Maudsley Biomedical Research Centre, the Medical Research Council (UK), the National Institute for Health Research, Horizon 2020, and the Innovative Medicines Initiative (European Commission). TC has received research grant support from the Medical Research Council (UK), the National Institute for Health Research, Horizon 2020 and the Innovative Medicines Initiative (European Commission), MQ, Autistica, FP7 (European Commission), the Charles Hawkins Fund, and the Waterloo Foundation.

## Author contributions

C.E., C.M.P., G.M., N.P., E.L., F.D., T.C., T.Bo., C.B., J.K.B., A.G.C., and D.G.G. contributed to the concept and design of the study. C.E., C.M.P., J.L., L.M.B., C.G., H.S., A.G.C., and D.G.M. analysed and interpreted the data. C.E. drafted the manuscript. J.L., L.M.B., C.G., and H.S. contributed to preprocessing and quality assessment of the data, and to the analytical pipeline. C.E. and B.O. EU-AIMS LEAP Group, D.G.M. contributed to the acquisition of the data. C.E., C.M.P., J.L., L.M.B., C.G., H.S., G.M., N.P., E.L., L.M., B.O., A.G.C., C.M.F., and D.G.M. provided administrative and technical support. C.E., G.M., E.L., F.D., T.C., T.Bo., C.B., J.K.B., C.A., T.ba., and D.G.M. obtained funding. E.H., D.K.B., M.S., J.R., and A.R. provided expert advice on clinical and pharmacological aspects. All authors contributed to the critical review of the manuscript.

## Funding

## Competing interests

J.K.B. has been in the past three years a consultant to/member of advisory board of/and/or speaker for Janssen Cilag BV, Takeda/Shire, Roche, Novartis, Medice, Angelini, and Servier. He is not an employee of any of these companies and not a stock shareholder of any of these companies. He has no other financial or material support, including expert testimony, patents, and royalties. TBa served in an advisory or consultancy role for AGB pharma, eye level, Infectopharm, Medice, Neurim Pharmaceuticals, Oberberg GmbH and Takeda. He received conference support or speaker's fee by AGB pharma, Janssen-Cilag, Medice and Takeda. He received royalties from Hogrefe, Kohlhammer, CIP Medien, Oxford University Press; the present work is unrelated to these relationships. T.C. has received royalties from Sage Publications and Guilford Publications. J.R. has received speaker's honoraria from Janssen, Hexal, Neuraxpharm and Novartis. D.G.M. has received consultancy fees from Roche and Servier. A.R. has received honoraria for lectures and/or advisory boards from Janssen, Boehringer Ingelheim, Compass, GH Research, SAGE/Biogen, LivaNova, Medice, Shire/Takeda, Newron, MSD, AbbVie, cyclerion. Also, he has received research grants from Medice and Janssen. The remaining authors declare no competing interests.

## Additional information

[1]Department of Child and Adolescent Psychiatry, University Hospital of the Goethe University, Frankfurt am Main, Germany. [2]Department of Forensic and Neurodevelopmental Sciences, Institute of Psychiatry, Psychology, and Neuroscience, King's College London, London, UK. [3]Cooperative Brain Imaging Center (CoBIC), Goethe University Frankfurt, Frankfurt am Main, Germany. [4]Department of Biosciences, Goethe University Frankfurt, Frankfurt am Main, Germany. [5]Department of Psychology, Institute of Psychiatry, Psychology, and Neuroscience, King's College London, London, UK. [6]Human Genetics and Cognitive Functions Unit, Institut Pasteur, University de Paris, Paris, France. [7]Department of Medical Neuroscience, Donders Institute for Brain, Cognition and Behaviour, Radboud University Medical Centre, Nijmegen, Netherlands. [8]Department of Child and Adolescent Psychiatry, Institute of Psychiatry and Mental Health Hospital General Univesitario Gregorio Maranon, and School of Medicine, Universidad Complutense de Madrid, CIBERSAM, Madrid, Spain. [9]Department of Child and Adolescent Psychiatry, Central Institute of Mental Health, Medical Faculty Mannheim, University of Heidelberg, Mannheim, Germany; German Center for Mental Health (DZPG), partner site Mannheim-Heidelberg-Ulm, Ulm, Germany. [10]Institute for Neuroradiology, University Hospital of the Goethe University, Frankfurt am Main, Germany. [11]Institute of Anatomy, University Medical Center of the Johannes Gutenberg-University, Mainz, Germany. [12]Department of Psychiatry, Psychosomatics, and Psychotherapy, University Hospital of the Goethe University Frankfurt, Frankfurt am Main, Germany. [13]Fraunhofer Institute for Translational Medicine and Pharmacology ITMP, Theodor-Stern-Kai 7, Frankfurt am Main 60596, Germany. [14]Institute for Translational Psychiatry, University of Münster, Münster, Germany. ✉e-mail: ecker@med.uni-frankfurt.de

