## [Transparent Peer Review file · Nature Communications]

Transcriptomic decoding of surface-based imaging phenotypes and its application to pharmacotranscriptomics

Corresponding Author: Professor Christine Ecker

Version 0:

Reviewer comments:

Reviewer #1

(Remarks to the Author)

Review: Transcriptomic decoding of surface-based imaging phenotypes – a new avenue for in vivo pharmacogenomics

In this paper by Ecker and colleagues, an exploratory analysis of imaging transcriptomics using the AHBA data and publicly available PET data is carried out mainly focusing on the GABA system and the serotonin system. The authors claim to be able to dissect two discrete classes of GABAA receptor units (alpha1 and alpha2), and that these expression patterns are related to specific behavioral symptoms and traits. Furthermore, the authors argue that their presented method is novel and represents a 'new avenue for advancing in vivo pharmacogenomics, and to guide the future development of targeted pharmacotherapies and personalized interventions.'

While it is indeed important work to investigate the association between gene expression and protein levels, and how these may link to differences in behavior and traits, there are a number of methodological concerns that may limit the conclusions of this work, such as how the correction for multiple comparisons was carried out for phenotype analysis. Furthermore, much of the work presented in this paper (also presented as novel) has already been carried out in previous papers, namely <https://pubmed.ncbi.nlm.nih.gov/29723639/>, <https://pubmed.ncbi.nlm.nih.gov/36209794/>, <https://www.nature.com/articles/s42003-019-0413-7>, <https://pubmed.ncbi.nlm.nih.gov/33610745/>. Lastly, to be clear, there are no predictive analyses carried out in this work, so all instances of prediction or decoding should be replaced with association or correlation (<https://pubmed.ncbi.nlm.nih.gov/31774490/>).

Major comments:

1. One of the key findings of this paper is that two differential patterns of alpha1 and alpha2 exist, and it is suggested that these are linked to two different forms of BZ receptors, namely BZ1 and BZ2. However, as shown in Hansen et al. 2022, there are no robust nor significant association between alpha1/alpha2 and BZ receptor levels. It is therefore not clear to this reviewer how potential differences in GABA gene expression would mechanistically be linked to potential differences in cortical thickness and behavior in developing individuals?
2. For the cohort of 279 subjects, it would be informative to know more about the population such as medication status and disease onset, or if subjects are suffering from one or more disorders. Furthermore, summary statistics for the various rating scales would be helpful to evaluate the presented results. Please report all subject-specific information and summary statistics to allow for evaluation of potential confounds.
3. Similar work on high-resolution gene-expression maps was published by Gryglewski and colleagues was published in 2018, so it would be useful to evaluate and discuss how the results from Gryglewski et al. 2018 may align or deviate from the data presented here. The data from Gryglewski are also publicly available, and so it would increase the robustness of this work, if a comparison to this data was made.
4. For the gene expression analysis, the gene expression has been normalized to be between 0 and 1. However, by normalizing the data, the contribution/weight of each subunit is removed, and will change the interpretations of the data. For example, as shown in both Sequeira et al. 2019 and Norgaard et al. 2019, the variance explained for the different GABA subunits is widely different, and this should be considered when evaluating the results.
5. It is not clear to this reviewer, what the serotonin part of the analysis is contributing to? All this work has been carried out before in previous papers, and it is not linked in any way to the phenotypic analysis? I would recommend the authors to delete this part of the paper, and instead focus on the GABA part.
6. The results from the phenotype analysis are currently only evaluated on the entire data, however, it would improve the reliability and robustness of these results, if the clustering and subsequent correlation to rating scales could be reframed to a

machine learning framework, where cross-validation can be used to evaluate the predictive performance. For more on this topic please see <https://pubmed.ncbi.nlm.nih.gov/31774490/>.

7. The gene expression data from AHBA mainly includes an elderly population and the PET data largely includes data from subjects in the range 20-40. These authors should comment/discuss how these differences in relation to the N=279 cohort might impact their results.

Minor comments:

1. Page 7: It is stated that 'The 5-HT_{2A}R gene (HTR2A) did not survive abagen quality assessments and was therefore also excluded from the analyses.', however, 5HT2A is included. Did you mean 5-HT_{1B} instead?
2. It is not clear where the PET data was obtained from. Was this from the original data (<https://xtra.nru.dk/BZR-atlas/>, <https://xtra.nru.dk/FS5ht-atlas/>) or from the Hansen et al. 2022, Nature Neuroscience paper? These data are not exactly the same, so it should be made clear in the paper, where the data originates from.
3. Please, if possible, make both derived data and code available to reproduce the results. This will largely increase the value of this work.

Reviewer #2

(Remarks to the Author)

In "Transcriptomic decoding of surface-based imaging phenotypes - a new avenue for in vivo pharmacogenomics", Ecker et al ask how gene transcription profiles should be compared with imaging-derived brain maps on the cortical surface. The authors first generate spatially-dense (vertex-level) gene expression maps from the AHBA, then compare three methods of comparing gene expression and imaging phenotypes (spatial correlation with autocorrelation-preserving null, linear mixed effects model, and general least squares model). They compare these methods by testing correlations between gene expression and receptor densities (5-HT and GABA_A). Since GABA_A is coded by multiple (19) genes, they consider clusters of GABA_A genes and how cortical thickness in a large population of people are aligned with either cluster of GABA_A gene expression profiles. Finally, they report that people more aligned with one vs the other GABA_A gene cluster show differences in behavior.

[1] The authors emphasize a novel method of relating gene expression with imaging-derived brain phenotypes. However, the methods that they test are not novel and the method that they converge on is a correlation with a spatial autocorrelation preserving null model, by now frequently employed in the field, including for relating imaging phenotypes with gene expression (e.g. Burt et al 2018 Nat Neurosci, Hansen et al 2022 NeuroImage, Murgas et al 2022 NeuroImage). Therefore I recommend removing claims of novelty in the manuscript and title.

[2] The authors reduce the dimensionality of the gene expression data by considering the first 9 principal components and generate spatial autocorrelation-preserving surrogate maps (via variogram matching) for these 9 gradients. I don't fully understand how these surrogate maps are being used in subsequent analyses.

For example, in Fig 4c, if I understand correctly then the "alpha-null decoding approach" is the Pearson's correlation between all genes and the 5-HT_{1a} receptor map, in which case the surrogates (and null models in general) do not play a part? (If this is the case, then stating "Pearson's correlation" would be much more clear than "alpha-null decoding approach".)

Similarly, in Fig. 5d, in the alpha-null row, are the authors assessing statistical significance using the surrogates from the gradients or using new surrogates from the GABA expression maps? Surrogates from the gradients would not be an appropriate null because the spatial autocorrelation of the gradients is different from the spatial autocorrelation of the expression map. Furthermore, if I understand correctly, the motivation for generating surrogates from the 9 gene expression gradients is due to the computational burden of generating surrogates for tens of thousands of genes. Why not generate surrogates for the imaging phenotype, which only needs to happen once instead of for each gene? In general, clarification on how the gradient surrogates are being used for significance testing would be helpful as I am not convinced that gradient surrogates are an appropriate null for a correlation between a specific gene's expression and an imaging phenotype.

[3] Relatedly, since the present analyses are conducted at the level of the cortical surface only, why did the authors use variogram-matching nulls rather than spatial permutation nulls (e.g. Alexander-Bloch et al 2018 NeuroImage)? Spatial permutation nulls will perfectly preserve the autocorrelation and can be implemented much faster, potentially forgoing the gradient surrogates (Markello & Misisic 2021 NeuroImage). This is because it isn't necessary to generate 1000+ surrogates for each brain map, rather, one could generate 1000+ rotations of vertices (that is, you would derive a vector of indices representing how to randomly permute your brain map while preserving spatial autocorrelation), regardless of the brain map, that can be reused in the permutation test no matter the correlation.

[4] How do dense gene expression maps generated using the present method compare with dense gene expression maps generated using the method from Wagstyl et al 2023 (MAGICC)?

[5] In Fig. 6b, 2/6 tests are statistically significant, of which one is statistically significant when parents report on their child's depression but not when the child themselves reports. As a result I recommend tampering some claims for example line 320 "we were able to dissect their putative functional role in vivo".

[6] I found it interesting that genes coding for different GABA_A subunits cluster into two groups with distinct cortical

expression profiles. While density distributions of different GABA_A receptors likely reflects unique neural signaling, why do the authors expect that GABA_A expression profiles would manifest as differences in cortical thickness? More justification of the link between cortical thickness and GABA_A type expression would be helpful.

[7] In the “alpha-null decoding” analysis, at what point was gene expression combined (averaged?) across donors, if at all?

[8] What is the differential stability of genes at the vertex-level versus their differential stability prior to the interpolation, or in a parcellated space? Does smoothing cause genes to become more similar to each other across donors?

[9] Since AHBA samples exist throughout the subcortex/brainstem/cerebellum and the two PET datasets used include binding throughout the whole brain, could the authors extend the analyses to non-cortical regions? (Except the cortical thickness analysis in Fig.6).

[10] The authors state in line 513 that “conventional (i.e. parametric) approaches are likely to overestimate the statistical relationship between two surface maps, thus inflating the number of false positives”. Note that it is not the fact that these conventional approaches are parametric that pose a problem but rather that conventional approaches typically assume that observations (in this case, brain regions) are exchangeable and statistically independent from one another. Indeed the spatial autocorrelation-preserving null employed here (variogram-matching) is a parametric approach.

[11] I suggest removing the statement in line 298 that a p-value between 0.05 and 0.1 is “marginally significant”.

[12] Typo in 242, it should read beta_{1,3} not 1-3.

[13] Typo in Fig 2b right, surrogate null is Burt et al 2020 not Burton.

Reviewer #3

(Remarks to the Author)

I approached the review of Dr. Ecker and colleagues' paper titled 'Transcriptomic decoding of surface-based imaging phenotypes - a new avenue for in vivo pharmacogenomics' with great enthusiasm. This work presents an advancement in imaging transcriptomics methodology aimed at broadening its application in pharmacogenomics. The authors successfully accomplished two main objectives: firstly, they developed an innovative approach for decoding transcriptomic patterns of high-resolution surface-based imaging data using spatially-dense representations of the human brain transcriptome generated from the AHBA. Secondly, they demonstrated how patterns of gene expression related to neurotransmission could manifest as different symptoms and traits mediated by structural neuroimaging.

The paper is commendably well-written and thorough in its presentation.

Regarding methodology, however, the interpolation of gene data to generate continuous transcriptomic maps introduces inherent risks and relies heavily on approximations. It is noted that the sample measurements from the AHBA are limited to few thousands across all donors and hemispheres, while individual transcript maps are defined for 40,962, indicating that over 80% of the data is mathematically inferred.

Furthermore, the validation presented in the paper is somewhat limited. While PET imaging has been commonly employed to validate the performance of imaging transcriptomics approaches, this paper only utilizes a selection of available PET templates for its analysis. Particularly, given the particular focus of the paper on the GABAergic system, the authors should make the effort to include other PET radiotracers (e.g. opioids, metabolic, endocannabinoids, NMDAR, etc). I'm surprised that previous work done within the same institution has not been leveraged for the validation of the proposed method (<https://www.nature.com/articles/s42003-022-03268-1>).

Lastly, the paper falls short in demonstrating its application to pharmacogenomics. Despite its title suggesting a connection, it does not incorporate any genomics (=DNA not mRNA) information, which seems speculative. Here, the suggestion would be to validate the proposed method with pharmacological imaging data using drugs with clear target affinity profiles (including GABAergic ones).

Overall, while the paper makes significant strides in advancing imaging transcriptomics methodology, there are areas where further validation and clarification are needed, particularly regarding the validation of the method and its application to pharmacogenomics

Reviewer #4

(Remarks to the Author)

The authors develop an approach to relate spatial transcriptomic profiles to in vivo brain imaging, and suggest that the method is valid by showing spatial association with benzodiazepine binding sites (from PET imaging) and associations with cortical thickness (with MRI) and its relation with clinical scores. The paper is remarkably interesting, and showcases a high level of sophistication by the authors, although in my opinion a few items could be improved:

A) The paper is remarkably long. It may not seem at first because the font used saves space, but it's a lengthy paper. The main text could probably be shortened by half, by removing information that, while very interesting as a scholarly overview, is not relevant for (and distract from) the main points.

- The Introduction, for example, has 3 long paragraphs; the first two could be replaced by just 1 of the size of the current first. We only start to understand what the paper is about when reaching the 3rd paragraph.
- The sections that start in lines 119 and 137 recapitulate the Methods section, again contributing to make the paper too lengthy. Even information on secondary analyses with LME and GLS are repeated here. The authors indicate these other analyses are less optimal, so why distract the main text with them?
- Also, plenty of information here is absent from its natural location, the Methods. Consider integrating such details there (e.g., samples, instruments), and shortening these sections by a lot, or dropping them completely from the main text.
- Another example is the lengthy paragraph detailing PCA in the Methods. PCA is very standard, one shouldn't need more than one or two line to indicate it was used and how.

B) While various steps are well detailed, the motivation for each one is not always obvious. Other details lack clarity. For example:

- Why does it matter whether probes have a valid Entrez ID?
- Early in the Methods we learn about fsaverage6 as template and that samples would be assigned to it. Further down the samples are (again?) mapped to fsaverage6. Where the two fsaverage6-related steps, or just one?
- What are the 1670 samples mentioned in line 459? Samples of what? There are only 6 brains... are these spatial locations?
- Why are coordinates only "roughly" symmetric (line 480). It should be possible to create perfectly symmetric templates by averaging coordinates in both sides. How "rough" is the residual asymmetry?
- Why are there duplicate vertices? (line 482)
- Is it really necessary to do kriging to upsample data? If those 1670 samples represent spatial locations in the cortex, wouldn't it be more faithful to the data to downsample the in vivo imaging to that resolution, like the authors do for their second analysis that starts in line 563? It would also dispense with the need to do PCA and the need to deal with loading to interpret results, while still using BrainSMASH for inference.
- What does the k_{nn} parameter mean? Why did it have to vary?
- Correction for multiple testing as described up to line 558 is ok. But maxT across PCs does not seem right, but in any case, very little details and no references are provided. What exactly was done here?
- If the method described between lines 562 and 577 does not take spatial autocorrelation into account, why use it? Why not just go straight with the GLS model?
- Why z-scoring cortical thickness was needed? The text says "to make individuals comparable" (lines 631-632) but that is too vague. Z-scoring on a per-vertex basis, done across the 279 subjects, surely alters the relative differences between neighboring vertices, and certainly impacts the spatial correlations with the expression maps. Shouldn't the raw cortical thickness values have been used instead when defining the clusters of subjects/strata?
- In line 223 we learn about a tool the authors seem to have used but that isn't described in the Methods (Protein Atlas). Please, give details of the role of this tool in the Methods.

C) Minor:

- The font used is too small even at size 11! Consider using the default fonts in MS Word to help us, poor-sighted reviewers! :-)
- Be judicious in the use of the word "prediction", here sometimes used to refer to the result of kriging, sometimes as the dependent variable in a statistical model. Currently, given the huge popularity of machine learning, the term "prediction" gained a new meaning that is not how it's used here.
- Same for "decoding". It isn't clear anything is being "decoded" in the manuscript...
- The paper hedges its bets a lot. Everything is "putative" (the word appears 11 times). If you can say something, and have supportive evidence, please say so.
- The hierarchical clustering (starting from line 607) depends only on gene expression data from the AHBA, right? If so, consider swapping the order of presentation with the previous subsection on PET data (that starts from line 587). The reason is just for clarity, in case the hierarchical clustering does not depend on the PET analysis (as it seems).
- "bifurcated" -> bifurcated (line 268)

All and all, I find the manuscript sound on its substance, and provides a valid, useful resource for people interested in using the Allen Brain data with in vivo imaging. But the paper is needlessly wordy, with many distractions that could (should in my opinion) be reduced.

Reviewer #5

(Remarks to the Author)

Thank you for the opportunity to review your manuscript. I appreciate the chance to engage with your work and to provide feedback.

Linking gene expression data with imaging-derived phenotypes to explore functional mechanisms is a promising approach in imaging transcriptomics. I commend your validation against PET data and the subsequent analysis of cortical expression patterns of GABAA-receptor subunits. However, I believe that the statements about the association between GABAA-receptor subunits and specific behavioral traits need further clarification and thus a major revision of the manuscript. My main concern relates to the methodology used to elaborate on how distinct cortical expression patterns relate to specific behavioral symptoms:

- Even though the approach to guide targeted drug discovery by linking gene expression patterns to in-vivo symptoms is promising, statements such as "Yet, proof-of-concept that neurotransmission-related patterns of gene expression predict specific symptoms and traits, and so could inform targeted (i.e., behaviorally guided) drug discovery, remains missing." (lines 99-101) may be too far-fetched at this stage.
- The authors repeatedly write about transcriptomic characteristics of individuals, suggesting that data of single subjects were linked with behavioral traits (e.g., lines 111-114: "Taken together, our study suggests that individuals can be stratified based on their transcriptomic association with two regionally distinct GABAAR subunit classes; one predominantly expressed in the limbic circuitry, and one with a regionally unspecific cortical expression landscape.") However, the analyses relate to the population-level as well as public atlas data, which limits the impact on "personalized interventions" or "personalized treatment strategies", which was stated in the introduction. To enhance the clarity of the manuscript, I recommend providing more detailed explanations of the underlying data basis. In particular, it should be stressed that the analyzed data sets are independent. In contrast, the information about the MR imaging data was provided rather late in the methods section. The authors should inform about the measures of cortical thickness by relocation of the corresponding paragraphs to an earlier part of the manuscript.
- In my opinion, the first two paragraphs of the results section (about "Spatially-dense (i.e., vertex-level) representations of the human brain transcriptome" and "Analytical framework for the transcriptomic decoding of high-resolution surface-based neuroimaging phenotypes") need shortening. There are too many details regarding the methodology that can be moved to the methods section.
- In general, multiple paragraphs of the manuscript (in the introduction, results, and discussion sections) seem redundant and should be condensed accordingly.
- The authors state that they "generated a high-resolution surface representation of the AHBA human brain transcriptome following a similar approach as described by Gryglewski et al. (2018) 19". In the discussion sections, the authors should discuss in more detail why they consider their approach superior to other previously published approaches.
- As a conclusion, the authors state: "Consequently, we demonstrated that individuals with delayed maturation in the limbic neurocircuitry, which was characterized by high expression of the $\alpha 2$ -containing GABAAR subunit Cluster 1, also had significantly higher levels of anxiety and depression than individuals falling into the $\alpha 1$ -containing GABAAR subunit Cluster 2, particularly during adulthood." This statement lacks clarity regarding the relationship between the analyzed (independent) data sets. Explaining the (statistical) association used to assume this (biological) relationship would enhance the comprehensibility and credibility of the findings.

Further remarks:

- In the introduction the outdated wording "typical and atypical brain organization" should be updated.
- The authors use the word "putative" several times in the introduction. I suggest refining the language by using alternative phrasings, avoiding the repetition of "putative".
- Lines 90-91: "For example, MR of GABA requires dedicated sequences, yet only report on 'bulk' GABA levels limiting interpretation 11." I believe, the authors wanted to write "MRS of GABA" ?
- Lines 102-104: "In the current study, we therefore (i) present a novel approach for the transcriptomic decoding of high-resolution surface-based IDPs or targets patterns using spatially-dense representations of the human brain transcriptome generated from the AHBA." I believe, the authors wanted to write "target patterns" ?
- Lines 115-117: "Based on these findings, we believe that the investigation of neurochemical pathways via fined-grained cartographic mapping of human gene expression offers promise for personalized medicine strategies, and represents a novel avenue to in vivo pharmacogenomics." I believe, the authors wanted to write "fine-grained" ?
- Lines 384-387: "Cortically, the expression of these subunits was tightly coupled with the regionally-unspecific expression pattern of genes in subunit cluster 2, suggesting that heteromeric GABAAR isoforms composed of $\alpha 1\beta 2\gamma 2$ subunits may play a crucial role in mediating a wide range of neurocognitive functions." This sentence is hard to understand and I suggest another wording.

Version 1:

Reviewer comments:

Reviewer #1

(Remarks to the Author)

While the authors have revised the manuscript to highlight and align more with previous research, the majority of the work presented in this manuscript has still been carried out before in previous papers (Gryglewski et al., Hansen et al. 2022, Norgaard et al. 2021, and Sequeira et al.), and therefore the novelty of this work is still limited, given the currently high priority in discussing these methods. Of novelty, is the section associating GABA patterns with CT/symptoms in an independent cohort, and the authors should therefore instead focus on these analyses and interpretations. The work presented prior to this could be added to the methods section, with references to the previous work developing these methods, and then describing any significant changes e.g. geodesic distance on the surface compared to euclidean. This is also what the authors state to be the goal of the work, as mentioned in the response letter, page 2 "Instead, we argue that cortical gene expression patterns as revealed by the AHBA implicate specific neural systems (i.e., cortical patterns), which in turn map onto specific behavioural symptoms and traits.". This is the novel part, and the authors should focus on this instead of presenting what they refer to as 'new methods', but as indicated by the authors also result in 'While the resulting surface-based gene expression signatures

are expected to be theoretically more accurate, as they account for the highly complex pattern of cortical folding, they were also highly correlated with the mRNA expression profiles published by Gryglewski et al. (2018). Additionally, the maps showed strong agreement with the spatially-smoothed dense expression patterns recently released by Wagstyl et al. (2023) on a lower-resolution (30k) surface template 19', from page 11 in the main manuscript, and 'Our finding also aligns with previous evidence suggesting that BZ binding sites can be categorized based on α subunit isoforms and their specific behavioral implications (reviewed in 13)' from page 14.

To my previous question #4 on taking into account the relative expression of each GABA subunit, the authors highlight in their answer my main concern namely differences in baseline expression between subunits, and large variation in expression across the cortex. However, it is still not clear from the manuscript and response from the authors how this was addressed, except stating that they focused on spatial correlations instead of covariances and there it was not relevant. But this does not alleviate the main concern of having different variances explained for the different subunits, and how this might impact these analyses and interpretations. For example, as shown by Norgaard et al. 2021 and Sequira et al. 2019, the two most abundant α -subunits are $\alpha 1$ and $\alpha 2$ with 16% and 11% variance explained, respectively. However, the remaining α -subunits only explain between 2-5% variance across all brain regions.

(Remarks on code availability)

This reviewer appreciates that code has been shared along with the manuscript.

Reviewer #2

(Remarks to the Author)

I thank the authors for responding to the reviewer comments in detail. The authors have clarified that they are using surrogate nulls of gradients to assess statistical significance of gene-IDP correlations. This is statistically incorrect. If the null hypothesis is that the correlation between a gene and an IDP are due to spatial autocorrelation, then the null needs to preserve the spatial autocorrelation of either the gene or the IDP. Gradients are not necessarily closely correlated with the gene of interest, does not necessarily have the same spatial autocorrelation, and represents a fundamentally different thing than the gene expression map itself. I appreciate that the authors ran the analysis making nulls from the IDP, and that they show that false positive rates are variable, but ultimately using the gradient as a null is an indirect method for assessing a relationship when there is an available direct method (generating surrogates on the IDP, using spins/surrogates/moran nulls/eigenstrapping on either phenotype, limiting computational burden by parcellating, etc). This method should be revised to apply the correct statistical null model.

Outside of this issue, the authors use appropriate methodology, the revised manuscript is much more clear and focused, and the figures are well made.

(Remarks on code availability)

Reviewer #3

(Remarks to the Author)

I would like to reiterate the positive feedback I provided in my previous review. The paper presents a compelling approach to linking transcriptomics with imaging phenotypes, and I appreciate the authors' efforts in addressing the methodological concerns raised earlier.

That said, while the revisions have improved many aspects of the paper, I remain somewhat unconvinced about the broad applicability of interpolating gene data to generate continuous transcriptomic maps for all types of targets. I recently encountered a paper (<https://pubmed.ncbi.nlm.nih.gov/33722642/>) that demonstrates how sensitive imaging associations can be when using different transcriptomics methods on the same data, which raises some questions about the generalizability of the approach presented here. However, for the systems of interest in this study—particularly GABA—the validation seems robust, and the methodology is clearly fit for purpose in this specific context.

A suggestion I have for further enhancing the paper is to consider providing access to other targets and genes beyond GABA and serotonin. This would allow for independent validation of the methodology and expand its applicability across different systems. Opening up this aspect would strengthen the paper's contribution to the field and potentially broaden its impact. Unfortunately I could not see this information in the shared code.

On a narrative level, the paper now reads more cohesively, and the findings are well-aligned with the overall conclusions. This work represents an exciting opportunity to advance our understanding of the connections between transcriptomic data and imaging phenotypes.

I have no further comments at this time.

(Remarks on code availability)

Reviewer #4

(Remarks to the Author)

I appreciate the authors responsiveness to the previous round of reviews. They have satisfactorily addressed all comments I had made.

Only one minor item: there is an important emphasis on speed, and I note that the extreme value method used for correction for multiple comparisons (here said to be similar to Westfall & Young, 1993) appears in fact remarkably similar to the "tail approximation" presented in <https://doi.org/10.1016/j.neuroimage.2016.05.068>. I think the latter reference is more informative about the procedure, which in itself seems correct to me.

(Remarks on code availability)

I applaud Nature journals for offering the possibility of reviewing code but a better option would be to allow a faster track for corrections or retractions when code errors are found after publication by users who really do try to use the code.

As a concrete example, Marek et al., Nature, 2022, has a coding mistake that invalidates its conclusions. The issue was flagged in a preprint by Italian researchers. Yet, somehow, the paper has received no expressions of concern, no corrections, and no "Matters Arising", despite the fact that Nature editors have been made aware of the problem.

That is something that needs to be fixed: allow fast track post-publication correction after code reviews by users worldwide. It is hard to rely on the already limited time available by a handful of reviewers who voluntarily review the manuscript itself.

Reviewer #5

(Remarks to the Author)

Dear authors,

thank you for submitting the revised manuscript "Transcriptomic decoding of surface-based imaging phenotypes and its potential for pharmacotranscriptomics".

I acknowledge the revisions made to the Introduction and Results sections, which now provide clearer and more accurate reflections of your approach and findings. The revised Discussion section now explains the importance of validating and translating these findings into clinical applications and highlights the necessity for future research.

Minor: I approve your revision after the initial feedback; however, I suggest to add a paragraph describing in more detail your decision to leave out subcortical regions from your analysis.

(Remarks on code availability)

Version 2:

Reviewer comments:

Reviewer #2

(Remarks to the Author)

I appreciate the authors' additional analysis comparing variograms of gradients and genes, as well as their added justification for using the gradient null approach rather than direct nulls. I have no further comments.

(Remarks on code availability)

We would like to thank all reviewers for taking the time to review our manuscript, and for their valuable and constructive suggestions on how its quality might be improved. We have now made extensive revisions, taking into account the reviewer's suggestions. In particular, we have shortened the manuscript significantly to remove redundancies and to highlight the two main objectives of the study, namely (1) the vertex-level decoding of IDPs, and (2) linking IDPs to molecular targets (e.g., GABA_ARs) to explore their functional role. Moreover, in response to the reviewer's suggestions, we have now implemented and assessed new decoding techniques, such as permutations and spin rotations of the target patterns. All software implementations are now freely available on github (see <https://github.com/christineecker/fsdecode>, and <https://github.com/christineecker/fsnulls>). Notably, the R project folder 'DecodeGABA' used for mapping IDPs to gene expression patterns contains sensitive information related to participant characteristics. We are therefore providing access to these files via a Dropbox download link:

<https://www.dropbox.com/scl/fo/8ih4wbtfouxhhg0q45y4i/AHWa5GkAprdyVPAKW7WEyg?rlkey=judzq0dyxodsx0d1f97mhx28w&dl=0>

The scripts used to generate Data Fig. 4 to 6 are contained within the 'analysis' folder. Last, during the revisions, we identified a minor mistake in the normalization of IDPs, resulting in slight differences in the results presented in Data Fig. 6b and 6d. We apologize for this oversight, which has now been corrected. However, this correction has not altered any of the main findings presented in the manuscript.

REVIEWER COMMENTS

Reviewer #1 (Remarks to the Author):

Review: Transcriptomic decoding of surface-based imaging phenotypes – a new avenue for in vivo pharmacogenomics

In this paper by Ecker and colleagues, an exploratory analysis of imaging transcriptomics using the AHBA data and publicly available PET data is carried out mainly focusing on the GABA system and the serotonin system. The authors claim to be able to dissect two discrete classes of GABAA receptor units (alpha1 and alpha2), and that these expression patterns are related to specific behavioral symptoms and traits. Furthermore, the authors argue that their presented method is novel and represents a 'new avenue for advancing in vivo pharmacogenomics, and to guide the future development of targeted pharmacotherapies and personalized interventions.' While it is indeed important work to investigate the association between gene expression and protein levels, and how these may link to differences in behavior and traits, there are a number of methodological concerns that may limit the conclusions of this work, such as how the correction for multiple comparisons was carried out for phenotype analysis. Furthermore, much of the work presented in this paper (also presented as novel) has already been carried out in previous papers, namely <https://pubmed.ncbi.nlm.nih.gov/29723639/>,

<https://pubmed.ncbi.nlm.nih.gov/36209794/>, <https://www.nature.com/articles/s42003-019-0413-7>, <https://pubmed.ncbi.nlm.nih.gov/33610745/>. Lastly, to be clear, there are no predictive analyses carried out in this work, so all instances of prediction or decoding should be replaced with association or correlation (<https://pubmed.ncbi.nlm.nih.gov/31774490/>).

Author reply: We appreciate the reviewer's comments, which we have addressed in detail in our replies to specific points. Most importantly, we have now implemented and evaluated different strategies for generating spatial null models, and explored their sensitivity and specificity in the context of gene expression decoding. This encompasses spatial permutation nulls of the target patterns, as suggested by Alexander-Bloch et al. (2018), in contrast to the variogram matching nulls that were precomputed for the nine gene co-expression gradients employed in permutation testing. We believe that these additional analyses are indeed crucial for assessing the validity and robustness of our gradient-based approach, particularly with respect to multiple comparison correction.

We also apologise if the reviewer felt that the novelty of our study was overstated. This has not been our intention, and we have fully acknowledged and referenced all previously published software tools and datasets, including the important prior works highlighted by the reviewer. We have therefore now carefully revised the manuscript to provide a more balanced presentation of our findings and to more accurately position our contributions within the context of existing research. Last, we have revised the manuscript to highlight the important distinction between in-sample estimates and out-of-sample predictions, which was a point that was also raised by reviewer #4. We agree that these are crucial issues, which we have carefully considered and addressed in our revisions.

Major comments:

1. One of the key findings of this paper is that two differential patterns of alpha1 and alpha2 exist, and it is suggested that these are linked to two different forms of BZ receptors, namely BZ1 and BZ2. However, as shown in Hansen et al. 2022, there are no robust nor significant association between alpha1/alpha2 and BZ receptor levels. It is therefore not clear to this reviewer how potential differences in GABA gene expression would mechanistically be linked to potential differences in cortical thickness and behavior in developing individuals?

Author reply: We fully recognise that gene expression patterns differ widely across the cortex and that there is significant inter-individual variability in the precise mapping between gene expression and protein binding. As we did not compare gene expression and PET data within the same set of individuals, our study design was not suited to quantify the association between gene expression and protein levels directly. As noted by the reviewer, this has been the research objective of other seminal studies such as Hansen et al., Norgaard et al., and Sequeira et al.. Instead, we argue that cortical gene expression patterns as revealed by the AHBA implicate specific neural systems (i.e., cortical patterns), which in turn map onto specific behavioural symptoms and traits. Thus, rather than proposing that differences in gene expression are

causative for neuroanatomical variation (e.g., differences in CT), we argue that examining the behavioural profiles of individuals with neuroanatomical differences that coincide with regions with low/high levels of expression can provide valuable insights into the functional role of a gene, and hence the behavioural consequences when targeted. Our approach is therefore purely correlational, i.e., based on the degree of spatial correlation between IDPs and gene expression patterns, which precludes any sort of causal inference.

Notably, within this framework, characterizing IDPs based on measures of CT represents only one approach for associating individuals with a candidate gene or genetic pathways. We examined CT because CT variability has previously been linked to genetic variation, including in GABA-ergic genes (Hollestein et al., 2023), as well as patterns of gene expression (see manuscript for details). Alternatively, one could examine other morphometric features, or even functional IDPs, to derive a spatial pattern that is reflective of individual variation. This is possible as our approach purely relies on the degree of spatial co-variation rather than a causal or mechanistic associations. Nevertheless, we agree with the reviewer that future studies should aim to elucidate the causative mechanisms behind variability in gene expression, protein density, neuroanatomical variation, and their link to behavioural variation. This has now been emphasized as an important limitation and a direction for future research in the Discussion section.

2. For the cohort of 279 subjects, it would be informative to know more about the population such as medication status and disease onset, or if subjects are suffering from one or more disorders. Furthermore, summary statistics for the various rating scales would be helpful to evaluate the presented results. Please report all subject-specific information and summary statistics to allow for evaluation of potential confounds.

Author reply: We much appreciate the opportunity to provide further details on participant demographics, mental health and medication status, as well as summary statistics for the main rating scales/questionnaires utilized in the present study. We have now added this information in Extended Data Tables 1 to 3.

3. Similar work on high-resolution gene-expression maps was published by Gryglewski and colleagues was published in 2018, so it would be useful to evaluate and discuss how the results from Gryglewski et al. 2018 may align or deviate from the data presented here. The data from Gryglewski are also publicly available, and so it would increase the robustness of this work, if a comparison to this data was made.

Author reply: This is an important comment. We consider the work by Gryglewski et al. (2018) to be absolutely foundational to the field of imaging transcriptomics, as Gryglewski et al. were the first to provide gene expression profiles across the entire cortical surface, at a resolution that is comparable to conventional surface-based IDPs. It was only our prior expertise in geodesic

distance mapping (e.g., Ecker et al. (2013)), which prompted us to extend this work and to derive whole-cortex mRNA signatures based on the geodesic rather than Euclidean vertex neighborhood for spatial interpolation. We are, however, pleased to report that the resulting patterns are highly correlated with the gene expression signatures released by Gryglewski et al., with spatial correlation coefficients exceeding a value > 0.9 for the target patterns examined in the current study. Moreover, our maps are highly similar to the patterns recently released by Wagstyl et al. (2023) on a lower-resolution (30k) surface template (see comment below). The results of these analyses have been included to the manuscript (see Extended Data Fig. 2 and 3).

4. For the gene expression analysis, the gene expression has been normalized to be between 0 and 1. However, by normalizing the data, the contribution/weight of each subunit is removed, and will change the interpretations of the data. For example, as shown in both Sequeira et al. 2019 and Norgaard et al. 2019, the variance explained for the different GABA subunits is widely different, and this should be considered when evaluating the results.

Author reply: All AHBA gene expression data was normalised to zero mean and unit variance within the *abagen* toolbox to make gene expression signatures comparable across genes and donors. Moreover, this avoids that the contribution or weight of any particular gene is inflated by the fact that this gene has a high baseline expression level, or a large variability in gene expression level across the cortex, relative to other genes; e.g., in case of the PCA. Notably, as we focused on spatial correlations rather than covariances (which is scale dependent), we can confirm that normalizing the size/magnitude of expression levels did not affect our reported results.

5. It is not clear to this reviewer, what the serotonin part of the analysis is contributing to? All this work has been carried out before in previous papers, and it is not linked in any way to the phenotypic analysis? I would recommend the authors to delete this part of the paper, and instead focus on the GABA part.

Author reply: We appreciate the reviewer's suggestion to remove the analysis of the 5-HTRs and to focus on GABA-Rs exclusively. However, our study had two main objectives: (i) present a methodological framework for the vertex-level decoding of IDPs, and (ii) to utilise the derived vertex-level gene expression patterns to link IDPs and associated clinical/behavioural phenotypes to molecular targets (and vice versa). To assess the sensitivity and specificity of our approach, we initially focused on serotonergic receptors with a particular emphasis on 5-HT_{1A}R. This receptor was chosen due to its relatively simple monogenic structure and its previously demonstrated strong correspondence between gene expression and protein binding (see Manuscript for details). We then applied our approach to GABA_ARs to demonstrate its effectiveness for targets with a more complex subunit structure, thus providing a more comprehensive evaluation of techniques. This is especially important for future publications

that may wish to utilize our approach to examine the molecular mechanisms of more complex imaging patterns. In response to the reviewer's comment, we have revised the manuscript to place greater emphasis on the two primary aims of the study, with the goal of strengthening the narrative arc from methods development to application.

6. The results from the phenotype analysis are currently only evaluated on the entire data, however, it would improve the reliability and robustness of these results, if the clustering and subsequent correlation to rating scales could be reframed to a machine learning framework, where cross-validation can be used to evaluate the predictive performance. For more on this topic please see <https://pubmed.ncbi.nlm.nih.gov/31774490/>.

Author reply: We would like to thank the reviewer for this valuable suggestion and have now added alternative analytical strategies to further corroborate our findings.

We fully acknowledge the important distinction between 'association' and 'prediction', and we agree with Poldrack et al. that an in-sample model fit index should not be reported as evidence for its' predictive accuracy, and have now revised the working in our manuscript accordingly and substituted the term 'prediction' with 'association'. We also appreciate the reviewer's suggestion to frame our analyses within a machine learning (ML) context and to assess the model's generalization performance through out-of-sample validation. Developing a model to predict measures of anxiety (or depression) based on cluster membership would require two distinct optimization processes: (1) fitting the clustering algorithm and evaluating its generalization performance (i.e., classification problem), and (2) training a model to predict measures of anxiety based on cluster membership (i.e., regression problem). Here, large sample sizes would be required to ensure that model estimates are reliable for both prediction steps. In addition, an independent sample would be required for out-of-sample validation. The two-step approach is also only meaningful if the models are independent; that is, the classification model transforms the data in such a way that provides meaningful information for the regression model. Consequently, if there is no clear rationale for a hierarchical approach, the two optimization steps might be integrated into a single model that directly predicts measures of anxiety given the input data. Also, as Poldrack et al. pointed out, prediction analyses should not be performed with samples smaller than several hundred observations, thus restricting the type of analyses that can be performed with the current data.

Nevertheless, to address the reviewer's comment, we have now placed the prediction of anxiety measures within a ML framework using Linear Regression to ensure that our findings comparable to the correlation and regression analyses presented in the manuscript. Also, Linear Regression only has one tuning parameter (i.e., the intercept), making it more suitable for small datasets compared to models with multiple tuning parameters. Using a 75/25% split of the data (i.e., train and test set respectively), we initially fitted a model that predicted anxiety based on the matrix of spatial correlations between IDPs and the mRNA pattern of the 13 GABA_AR subunit genes using 3-fold cross validation in our adult sample of N=82 individuals. Moreover, we fitted a model predicting anxiety levels using the individual's accumulated degree of

neuroanatomical diversity within the cortical mask for the ‘limbic’ GABA_AR subunit *Cluster 1*. Both models provided, however, only moderate out-of-sample prediction accuracies, which may reflect of the relatively small sample size overall (N(train)=62, N(test)=20). We have included the results of these additional analyses in the Supplement (see Extended Data Fig. 12). Moreover, we have highlighted the future need of performing out-of-sample validations in the *Discussion* section.

7. The gene expression data from AHBA mainly includes an elderly population and the PET data largely includes data from subjects in the range 20-40. These authors should comment/discuss how these differences in relation to the N=279 cohort might impact their results.

Author reply: We appreciate this comment. As the reviewer pointed out, there was limited overlap between the age range of the AHBA donors (24-57 years), the PET atlas (20-40 years), and out participant sample (7-31 years). This may also help to explain why the brain-behavioural correlations presented in Figure 6 were only significant in adults, and not in children and adolescents. This important point/justification has now been mentioned in the *Discussion* section, where we discuss the behavioural correlations.

Minor comments:

1. Page 7: It is stated that ‘The 5-HT_{2A}R gene (HTR2A) did not survive abagen quality assessments and was therefore also excluded from the analyses.’, however, 5HT2A is included. Did you mean 5-HT_{1B}R instead?

Author reply: We apologize for this typo, which has now been corrected. The reviewer rightly notes that 5-HT_{1B}R was excluded and not 5-HT_{2A}R.

2. It is not clear where the PET data was obtained from. Was this from the original data (<https://xtra.nru.dk/BZR-atlas/>, <https://xtra.nru.dk/FS5ht-atlas/>) or from the Hansen et al. 2022, Nature Neuroscience paper? These data are not exactly the same, so it should be made clear in the paper, where the data originates from.

Author reply: Thank you for this comment. We can confirm that the PET data analysed in our study was obtained from the original source using the weblink provided by the reviewer, mapped onto the *fsaverage6* surface template. This has now been clarified in the *Methods*.

3. Please, if possible, make both derived data and code available to reproduce the results. This will largely increase the value of this work.

We are very happy to release all code via publicly available resources. This is also a requirement of the journal, and we had therefore provided a Dropbox link to all scripts/code accompanying the submission. This should have been forwarded to all reviewers alongside the submission. Note, we decided to share the code via Dropbox rather than github because of the large amount of meta data required for the gene expression decoding, which relies on the N=1,000 pre-computed spatial nulls for each expression gradient. However, the `fsnulls` and `fsdecode` R packages that use `git annex` and `datalad` are now available on github, as outlined above.

Reviewer #2 (Remarks to the Author):

In “Transcriptomic decoding of surface-based imaging phenotypes - a new avenue for in vivo pharmacogenomics”, Ecker et al ask how gene transcription profiles should be compared with imaging-derived brain maps on the cortical surface. The authors first generate spatially-dense (vertex-level) gene expression maps from the AHBA, then compare three methods of comparing gene expression and imaging phenotypes (spatial correlation with autocorrelation-preserving null, linear mixed effects model, and general least squares model). They compare these methods by testing correlations between gene expression and receptor densities (5-HT and GABAa). Since GABAa is coded by multiple (19) genes, they consider clusters of GABAa genes and how cortical thickness in a large population of people are aligned with either cluster of GABAa gene expression profiles. Finally, they report that people more aligned with one vs the other GABAa gene cluster show differences in behavior.

[1] The authors emphasize a novel method of relating gene expression with imaging-derived brain phenotypes. However, the methods that they test are not novel and the method that they converge on is a correlation with a spatial autocorrelation preserving null model, by now frequently employed in the field, including for relating imaging phenotypes with gene expression (e.g. Burt et al 2018 Nat Neurosci, Hansen et al 2022 NeuroImage, Murgas et al 2022 NeuroImage). Therefore I recommend removing claims of novelty in the manuscript and title.

Author reply: We respectfully acknowledge the reviewer’s comment with regards to the novelty of the study, and fully acknowledge that our approach heavily utilizes previously published software tools such as the BrainSMASH toolbox developed by Burt et al. (2018). All of these tools and data utilized in our study were fully referenced. While these tools can be used to assess the statistical significance of spatial correlations, there is however currently no field-accepted framework for the vertex-level transcriptomic decoding of high-resolution surface-based IDPs using all genes within the AHBA, which our study aimed to implement and evaluate. Moreover, we consider the idea of aligning IDPs with the gene expression profiles of candidate targets to infer their functional role novel. In response to the reviewer’s comment, we have removed the term "novel" from all references throughout the revised manuscript, including the title.

[2] The authors reduce the dimensionality of the gene expression data by considering the first 9 principal components and generate spatial autocorrelation-preserving surrogate maps (via variogram matching) for these 9 gradients. I don’t fully understand how these surrogate maps are being used in subsequent analyses. For example, in Fig 4c, if I understand correctly then the “alpha-null decoding approach” is the Pearson’s correlation between all genes and the 5-HT1a receptor map, in which case the surrogates (and null models in general) do not play a part? (If this is the case, then stating “Pearson’s correlation” would be much more clear than

“alpha-null decoding approach”.) Similarly, in Fig. 5d, in the alpha-null row, are the authors assessing statistical significance using the surrogates from the gradients or using new surrogates from the GABA expression maps? Surrogates from the gradients would not be an appropriate null because the spatial autocorrelation of the gradients is different from the spatial autocorrelation of the expression map. Furthermore, if I understand correctly, the motivation for generating surrogates from the 9 gene expression gradients is due to the computational burden of generating surrogates for tens of thousands of genes. Why not generate surrogates for the imaging phenotype, which only needs to happen once instead of for each gene? In general, clarification on how the gradient surrogates are being used for significance testing would be helpful as I am not convinced that gradient surrogates are an appropriate null for a correlation between a specific gene’s expression and an imaging phenotype.

Author reply: These are all extremely valuable questions and suggestions, and we would like to thank the reviewer for asking us to explore alternative analytical approaches. As the reviewer rightly pointed out, the challenge in vertex-level decoding of IDPs does not lie in computing and assessing spatial correlations themselves, but rather in managing the high computational demands associated with decoding a large number of genes or IDPs. For this reason, we opted for a gradient-based approach with precomputed spatial nulls, enabling the decoding of IDPs in a relatively short amount of time on any standard PC or laptop.

For this purpose, we initially explored various spatial null models, including the spin model by Alexander-Bloch et al. (2018), and the computationally more expensive variogram-matching model by Burt et al. (2020). The spin model has many advantages: it is extremely fast to compute, and the same vector of rotated indices can easily be applied to a large number of surface overlays. Moreover, as the surface overlay is simply rotated across the surface, it has the same smoothness as the target pattern. However, in this model, the medial wall label is not masked out, but instead moves across the cortical surface. Thus, depending on the particular rotation, this can either inflate or deflate the spatial correlation, which we considered a drawback. In contrast, while the variogram-matching model by Burt et al. (2020) enables the user to mask the medial wall, it is so computationally expensive that it cannot easily be applied to permute a large number of IDPs or gene expression patterns. This is due to various regions; e.g., it relies on the geodesic distance matrix between all vertex pairs. This matrix can be precomputed for a surface mesh, but requires roughly 30GB of RAM for the *fsaverage6*. Moreover, the quality of the surrogate maps heavily relies on the *knn* parameter that has to be optimised for each IDP, ranging from the default value of 1,500, up to the total number of vertices per hemisphere (e.g., 41k). On a high performance MacPro desktop with > 200 GB of RAM, this step roughly takes between 3 to 5 hours to compute for a single IDP using 10 nodes.

Given these computational constraints, we opted for a gradient-based approach across nine co-expression patterns, utilizing precomputed variogram-matching nulls after optimizing the *knn* parameter. Nominal and adjusted permutation *p*-values were subsequently derived both within and across gradient surrogates: for each individual gene across the nulls of the gradient with which the gene is most strongly correlated, and across gradient nulls to account for multiple comparisons (see Figure 2).

In response to the reviewer's valuable comment and to further evaluate the robustness and validity of our approach, we have now explored two alternative decoding methods. These methods involve 1,000 permutations of the target patterns (e.g. 5-HT PET maps), using (1) Alexander-Bloch spin rotations, and (2) variogram-matching nulls according to Burt et al. (2020). These models were subsequently compared to the gradient-based approach. The results are presented in Extended Data Figures 7 to 9. In brief, the spin model was the least conservative, resulting in a large number of false positives, especially at less stringent statistical thresholds of $p_{adj} < 0.05$ and 0.01 . The variogram-matching null model applied to the target pattern proved to be overly stringent, identifying only a few genes as significantly associated with the target pattern. In comparison, the gradient-based alpha-null decoding approach offered a good trade-off between sensitivity and specificity across various levels of statistical stringency, and provided results that were comparable with generating surrogates of the target patterns. It therefore seems that gradient surrogates provide justifiable null models for the expression patterns of individual genes.

There are also significant differences in computation time between approaches. Decoding using null models of the target pattern takes roughly 8 minutes for a single IDP, in addition to the time needed to generate the null models (3-5 hours for Burt et al., 3 minutes for spin models). The gradient-based approach that provides comparable results only takes 2 minutes to run for a single IDP. It is therefore particularly well suited for decoding a large number of IDPs. All presented approaches have now been implemented in an R package called 'fsdecode', which can be freely accessed via *github* (<https://github.com/christineecker/fsdecode>).

[3] Relatedly, since the present analyses are conducted at the level of the cortical surface only, why did the authors use variogram-matching nulls rather than spatial permutation nulls (e.g. Alexander-Bloch et al 2018 NeuroImage)? Spatial permutation nulls will perfectly preserve the autocorrelation and can be implemented much faster, potentially forgoing the gradient surrogates (Markello & Misic 2021 NeuroImage). This is because it isn't necessary to generate 1000+ surrogates for each brain map, rather, one could generate 1000+ rotations of vertices (that is, you would derive a vector of indices representing how to randomly permute your brain map while preserving spatial autocorrelation), regardless of the brain map, that can be reused in the permutation test no matter the correlation.

Author reply: This is another valuable comment, which we have addressed in our response to the comment above. We agree that it would have been much easier to create surrogates for the target patterns (e.g., PET maps) rather than for individual gene expression patterns. However, given the high computational demands associated with optimizing the *knn* parameter, we opted for the gradient based permutation approach that has been described in further detail in the Methods section. The reasons for employing variogram-matching nulls rather than spatial permutation models are also explained above.

[4] How do dense gene expression maps generated using the present method compare with dense gene expression maps generated using the method from Wagstyl et al 2023 (MAGICC)?

Author reply: The multiscale atlas of gene expression provided by Wagstyl et al. (2024) is another excellent resource, which was published while we were working on our spatial interpolation model motivated by the work of Gryglewski et al. (2018). The gene expression patterns released by Greyglewski et al. have the same spatial resolution as the gene expression signatures derived in the present study (i.e., on the *fsaverage6* with 41k vertices). In response to reviewer #1, we have therefore now directly compared both approaches and present the comparison in Extended Data Fig. 2. The gene expression patterns provided by Wagstyl et al. characterise gene expression patterns at ~30k vertices, and it is therefore not easily possible to assess the spatial correlation between both patterns. In response to the reviewer's comment, we have included an additional figure (see Extended Data Fig. 3), which allows for a side-by-side visual comparison with the gene expression maps provided by the MAGICC toolbox.

[5] In Fig. 6b, 2/6 tests are statistically significant, of which one is statistically significant when parents report on their child's depression but not when the child themselves reports. As a result I recommend tampering some claims for example line 320 "we were able to dissect their putative functional role in vivo".

Author reply: In response to the reviewers' comments, we have now made extensive modifications to the Discussion section, which we hope reflect our findings more accurately. This sentence, among others, has now been revised within the broader context of the Discussion.

[6] I found it interesting that genes coding for different GABA_A subunits cluster into two groups with distinct cortical expression profiles. While density distributions of different GABA_A receptors likely reflects unique neural signaling, why do the authors expect that GABA_A expression profiles would manifest as differences in cortical thickness? More justification of the link between cortical thickness and GABA_A type expression would be helpful.

Author reply: Thank you for raising this important question, which was also brought up by reviewer #1. We therefore kindly refer the reviewer to our response to this comment on page 2.

[7] In the "alpha-null decoding" analysis, at what point was gene expression combined (averaged?) across donors, if at all?

Author reply: We thank the reviewer for this question. Prior to spatial interpolation, the AHBA samples from each donor were mapped to the nearest vertices on the FreeSurfer *fsaverage6* surface template using their x, y, z coordinates in volume space. In a few instances, samples from different donors were assigned to the same vertex. In such cases, the mRNA values at that vertex were averaged across donors. A similar approach was adopted by Gryglewski et al. (2018). However, since the data was previously normalized using the *abagen* toolbox, and this occurred in only a few instances, their impact is negligible. We have now added further clarification of our approach to the Methods section.

[8] What is the differential stability of genes at the vertex-level versus their differential stability prior to the interpolation, or in a parcellated space? Does smoothing cause genes to become more similar to each other across donors?

Author reply: To evaluate the impact of interpolation on gene correlations, we have now examined the spatial correlations between the GABA_AR subunit genes analyzed in this study before and after spatial interpolation. We find that while the spatial interpolation – and thus smoothing - increased the spatial correlation overall relative to the non-interpolated data, the positioning of genes relative to each other (i.e., their differential stability) remained stable. More specifically, the Spearman rank correlation between spatial correlations pre and post interpolation was 0.9. We thank the reviewer for raising this important point. The results of these additional analyses are now presented in Extended Data Fig. 10.

[9] Since AHBA samples exist throughout the subcortex/brainstem/cerebellum and the two PET datasets used include binding throughout the whole brain, could the authors extend the analyses to non-cortical regions? (Except the cortical thickness analysis in Fig.6).

Author reply: The reviewer rightly notes that the AHBA also provides gene expression data in subcortical brain regions, as well as the cerebellum. These are of particular interest for molecular targets with a predominantly subcortical pattern of expression, such as the dopaminergic system. In the current study, we focused on cortical patterns of expression exclusively as these are of particular interest to neuropsychiatric disorders. These are typically characterized by fine-grained neuroanatomical differences within large, spatially distributed cortical systems rather than subcortical brain regions. Furthermore, while techniques and software for region-based transcriptomic decoding are available, there are currently no established frameworks for decoding spatially dense vertex-level IDPs. This presents specific statistical challenges compared to region-based approaches, such as dealing with autocorrelation in spatially embedded signals. Techniques specifically developed for vertex-level decoding can therefore not easily be translated to the regional level.

Moreover, in vertex-level decoding, spatial correlations are computed across thousands of vertices (e.g., >41k) on the cortical surface. The number of subcortical brain regions (~6-10

per hemisphere) is therefore significantly lower than the number of vertices, and their relative impact on the spatial correlations would be minimal overall. In addition, as also noted by the reviewer, not every cortical feature is directly comparable across cortical and subcortical brain regions; e.g., CT is a cortical rather than subcortical measure of brain organization. Thus, in addition to the statistical considerations, limited biological insight might be gained from concatenating cortical and subcortical features.

[10] The authors state in line 513 that “conventional (i.e. parametric) approaches are likely to overestimate the statistical relationship between two surface maps, thus inflating the number of false positives”. Note that it is not the fact that these conventional approaches are parametric that pose a problem but rather that conventional approaches typically assume that observations (in this case, brain regions) are exchangeable and statistically independent from one another. Indeed the spatial autocorrelation-preserving null employed here (variogram-matching) is a parametric approach.

Author reply: This is correct and we have now amended the manuscript accordingly.

[11] I suggest removing the statement in line 298 that a a p-value between 0.05 and 0.1 is “marginally significant”.

Author reply: We appreciate the reviewer’s comment and have changed the text to highlight a non-significant finding.

[12] Typo in 242, it should read beta_{1, 3} not 1-3.

Author reply: We apologize for this typo, which has now been corrected. Thank you for bringing this to our attention.

[13] Typo in Fig 2b right, surrogate null is Burt et al 2020 not Burton.

Author reply: We apologize for this typo, which has now been corrected.

Reviewer #3 (Remarks to the Author):

I approached the review of Dr. Ecker and colleagues' paper titled 'Transcriptomic decoding of surface-based imaging phenotypes - a new avenue for in vivo pharmacogenomics' with great enthusiasm. This work presents an advancement in imaging transcriptomics methodology aimed at broadening its application in pharmacogenomics. The authors successfully accomplished two main objectives: firstly, they developed an innovative approach for decoding transcriptomic patterns of high-resolution surface-based imaging data using spatially-dense representations of the human brain transcriptome generated from the AHBA. Secondly, they demonstrated how patterns of gene expression related to neurotransmission could manifest as different symptoms and traits mediated by structural neuroimaging. The paper is commendably well-written and thorough in its presentation.

Regarding methodology, however, the interpolation of gene data to generate continuous transcriptomic maps introduces inherent risks and relies heavily on approximations. It is noted that the sample measurements from the AHBA are limited to few thousands across all donors and hemispheres, while individual transcript maps are defined for 40,962, indicating that over 80% of the data is mathematically inferred.

Author reply: We agree with the reviewer that the interpolation of the mRNA expression data to the spatial resolution of the surface-based neuroimaging data poses risks, and heavily relies on the assumptions and quality of the spatial interpolation procedure.

Due to the sparsity of the AHBA data relative to the resolution of the surface-based IDPs, a significant proportion of the mRNA data is theoretically/mathematically inferred rather than directly measured. This prompts the question whether one should restrict the analysis to existing AHBA samples exclusively to minimise the probability of detecting spurious effects (also see reviewer #5). Notably, this issue typically occurs when combining two signals with vastly different resolutions, leading to either (1) a 'pseudo-inflation' of the lower-resolution signal when resampled into a higher dimension, or (2) a significant loss of information from the higher-resolution signal when resampled into a lower dimension. Consequently, downsampling high-resolution surface-based IDPs to match the native resolution of the AHBA would lead to a substantial loss of signal variation across the cortex. This is especially problematic for signals with high spatial frequency fluctuations across the cortical surface, which are commonly observed both in typically developing brains and in those affected by mental health conditions.

On the other hand, 'up-sampling' the AHBA data to the resolution of the IDPs may inflate the relationship between gene expression and imaging signatures. It is therefore crucial to assess how spatial interpolation impacts the correlations between genes and IDPs and to compare these results with those obtained using alternative methods that rely solely on existing AHBA samples, such as the LME and GLS approaches. Notably, we observe that although spatial interpolation generally increases spatial correlations, the relative ranking or positioning of

genes with respect to each other remains largely consistent across techniques (see Data Fig. 4c). Furthermore, to identify the differential stability of genes, we have now examined the spatial correlations between the GABA_AR subunit genes analyzed in this study before and after spatial interpolation. We find that while the spatial interpolation increased the spatial correlation overall relative to the non-interpolated data, the positioning of genes relative to each other remained stable (see Extended Data Fig. 11). We have also now examined alternative approaches for assessing the statistical relationship between spatially-interpolated high resolution gene expression patterns and surface-based target maps, demonstrating a high correspondance between significant genes across techniques (see Extended Data Fig. 7 to 9). This indicates that the differential stability of genes at the vertex level is approximately equivalent to their differential stability at the sample level, and that similar gene sets are highlighted both pre and post interpolation.

We believe these additional analyses substantiate our approach, and we are very grateful to the reviewers for giving us the opportunity to provide further justification. In light of our findings, we consider vertex-level decoding of IDPs using spatially interpolated mRNA expression patterns to be a valuable alternative to the conventional region/parcellation-based approach. Moreover, vertex-level decoding may offer advantages when dealing with IDPs characterized by high-spatial frequency signal fluctuations, which cannot be adequately captured by the native resolution of the AHBA.

Furthermore, the validation presented in the paper is somewhat limited. While PET imaging has been commonly employed to validate the performance of imaging transcriptomics approaches, this paper only utilizes a selection of available PET templates for its analysis. Particularly, given the particular focus of the paper on the GABAergic system, the authors should make the effort to include other PET radiotracers (e.g. opioids, metabolic, endocannabinoids, NMDAR, etc). I'm surprised that previous work done within the same institution has not been leveraged for the validation of the proposed method (<https://www.nature.com/articles/s42003-022-03268-1>).

Author reply: We greatly appreciate the reviewer's suggestion to consider using alternative PET templates or radiotracers to further validate our approach. However, our study was not designed to address the important question of the relationship between mRNA expression and protein density, a topic that has already been thoroughly investigated in seminal research by Hansen et al., involving a diverse range of PET targets. Rather, as previously highlighted, the PET data was primarily utilized to (1) evaluate the validity of the spatially interpolated mRNA expression patterns, and (2) serve as a 'true positive' reference for assessing the sensitivity and specificity of the statistical frameworks utilized for the transcriptomic decoding of surface-based IDPs.

In this context, the PET data used in our study had to meet certain criteria, and we would like to apologise for not outlining these more clearly in the original version of the manuscript. First, as our study focused on surface-based IDPs exclusively, it was important to examine molecular targets with a predominantly cortical pattern of expression and/or receptor density. This

excludes e.g., the dopaminergic system, which has a largely subcortical mechanism of action. Second, as the PET maps served as ‘true positive’ for evaluating the decoding approaches, we had to employ PET templates with a high correlation between mRNA expression and protein density, which has previously been demonstrated for the HTR_{1A} receptor in particular (see Manuscript for references). Third, to be able to discern separate cortical networks that tap into distinct functional domains, we had to identify a target with a sufficiently complex molecular structure, such as the GABA_A receptor. Here, previous evidence suggests that distinct isoforms are expressed in different regions of the brain, each associated with specific cognitive and behavioral domains. Fourth, we required PET data for which surface data was available. Although a large variety of PET targets are currently available, not all of them provide a high quality rendering onto the FreeSurfer *fsaverage6* surface template. Although these limitations may have constrained the scope (and therefore the novelty) of our findings, they enabled us to effectively address the primary objectives of our study.

Nonetheless, we agree with the reviewer that it would be highly beneficial to incorporate alternative tracers to further investigate the GABAergic system, including those suggested by the reviewer. While this would go beyond the scope of the current study, this is definitely something we aim to investigate in the future, also accommodating PET and structural MRI data acquired in the same set of individuals, and using a pharmacological challenge (also see comment below). We therefore thank the reviewer again for this valuable comment. In response, we have added further justification for the selection of PET maps to the *Methods* section and emphasize the need for future research in the *Discussion* section.

Lastly, the paper falls short in demonstrating its application to pharmacogenomics. Despite its title suggesting a connection, it does not incorporate any genomics (=DNA not mRNA) information, which seems speculative. Here, the suggestion would be to validate the proposed method with pharmacological imaging data using drugs with clear target affinity profiles (including GABAergic ones).

Author reply: We respectfully acknowledge the reviewer’s comment that additional research is needed to relate our transcriptomic findings to genomic variation.

We have now extensively revised the manuscript to highlight the two main objectives of the study; i.e., the statistical evaluation of vertex-level decoding techniques, and linking molecular targets to behavioural domains via spatial correlations between IDPs and candidate gene expression patterns. While our findings indicate that the spatial alignment between IDPs and target patterns implicates specific behavioural domains, the next steps will be to determine whether the statistical association between an IDP and a target patterns also aligns with PET and/or pharmacological imaging data acquired within the same set of individuals. Also, it will be important to determine which type of approach and/or data modality best predicts the individual’s response to pharmacological intervention; e.g., serve as an enrichment marker for clinical trials.

The reviewer also rightly notes that our study did not incorporate any genetic information, which may provide important novel insights into the specific genetic and transcriptomic mechanisms underpinning complex behavioural phenotypes and their pharmacological modulation. We have therefore now carefully reassessed all main claims regarding the implications and utility of our approach with respect to pharmacogenomics. We have also amended the title to more accurately reflect the content of our study. Last, we have highlighted these issues as important directions for future research in the *Discussion* section.

Overall, while the paper makes significant strides in advancing imaging transcriptomics methodology, there are areas where further validation and clarification are needed, particularly regarding the validation of the method and its application to pharmacogenomics.

Author reply: We thank the reviewer once again for their valuable comments and suggestions, and fully acknowledge the need for further validation and application of our approach in the clinical setting. This includes predicting an individual's response to targeted pharmacological intervention based on the spatial alignment between IDPs and gene expression patterns, which we have now emphasized as a crucial avenue for future research in the *Discussion* section.

Reviewer #4 (Remarks to the Author):

The authors develop an approach to relate spatial transcriptomic profiles to in vivo brain imaging, and suggest that the method is valid by showing spatial association with benzodiazepine binding sites (from PET imaging) and associations with cortical thickness (with MRI) and its relation with clinical scores. The paper is remarkably interesting, and showcases a high level of sophistication by the authors, although in my opinion a few items could be improved:

A) The paper is remarkably long. It may not seem at first because the font used saves space, but it's a lengthy paper. The main text could probably be shortened by half, by removing information that, while very interesting as a scholarly overview, is not relevant for (and distract from) the main points.

- The Introduction, for example, has 3 long paragraphs; the first two could be replaced by just 1 of the size of the current first. We only start to understand what the paper is about when reaching the 3rd paragraph.

- The sections that start in lines 119 and 137 recapitulate the Methods section, again contributing to make the paper too lengthy. Even information on secondary analyses with LME and GLS are repeated here. The authors indicate these other analyses are less optimal, so why distract the main text with them?

- Also, plenty of information here is absent from its natural location, the Methods. Consider integrating such details there (e.g., samples, instruments), and shortening these sections by a lot, or dropping them completely from the main text.

- Another example is the lengthy paragraph detailing PCA in the Methods. PCA is very standard, one shouldn't need more than one or two line to indicate it was used and how.

Author reply: We agree with the reviewer that some sections were overly lengthy and contained repetitions of the *Methods* section.

We have now shortened the manuscript significantly and made extensive revisions to the *Introduction*, *Methods*, and *Results* sections, taking into account the reviewer's comments. More specifically, we have rewritten the introduction to focus predominantly on the main objectives of the study, namely (1) the vertex-level decoding of IDPs, and (2) linking IDPs to molecular targets (e.g., GABA_ARs) to explore their functional role. We have also relocated methodological details from the *Results* section to the *Methods* section and shortened the *Methods* section itself; e.g., with regards to the PCA and the *abagen* preprocessing of the AHBA, which has been described in details previously. Finally, we have provided additional justification for using the LME and GLS techniques, which enabled us to evaluate the

robustness of the gradient-based spatial interpolation approach compared to methods based on the native resolution of the AHBA.

- B) While various steps are well detailed, the motivation for each one is not always obvious. Other details lack clarity. For example: Why does it matter whether probes have a valid Entrez ID?

Author reply: The assignment of genes to Entrez IDs is performed using the *abagen* toolbox, which was developed to standardize workflows in imaging transcriptomics.

In the original release of the AHBA data, genes names were provided according to the HGNC (HUGO Gene Nomenclature Committee) system; i.e., in ‘symbols’. While this system provides a standardized approach to naming genes, there are several disadvantages. For example, despite standardization, a single gene can have multiple names or ‘aliases’, and the same alias might refer to different genes. This makes it difficult to interpret findings across studies. Also, the HGNC regularly updates gene names, which means that genes are not consistently referenced over time. The *abagen* toolbox therefore re-annotates genes in the original AHBA release to the newest nomenclature. Because gene symbols can vary across databases, publications, or time, it is therefore advantageous to utilize a unique and stable identifier for each gene, such as Entrez IDs, which ensures the consistent referencing across studies and databases.

Moreover, Entrez IDs integrate various types of data such as gene function, expression, sequences, across-species comparison etc. into a single platform, which facilitates the interpretation of findings. There are various reasons why a gene does not have a valid Entrez ID; e.g., the gene is not officially recognized, the gene is a pseudogene or non-coding RNA, or the gene symbol is an alias or synonym. To avoid these issues, it is therefore recommended to exclude genes without a valid Entrez ID to ensure that genes are consistently referenced across studies, which makes findings comparable. Since the reannotation of AHBA gene symbols to the latest nomenclature is integrated within the *abagen* toolbox, which we have cited in the condensed Methods section, we now direct the reader to the original reference by Arnatkevičiūtė et al. (2019) for further clarification on this point.

- Early in the Methods we learn about fsaverage6 as template and that samples would be assigned to it. Further down the samples are (again?) mapped to fsaverage6. Where the two fsaverage6-related steps, or just one?

Author reply: We confirm that the *fsaverage6* template was used twice. First, it was employed within the *abagen* toolbox to exclude AHBA tissue samples from subcortical brain regions (e.g., thalamus, basal ganglia, brainstem, etc.) and the cerebellum, which were not included in the current study. In this context, we use the term ‘samples’ to refer to the actual tissue pieces collected from various regions of the donor brains (see also the comment below). Second, the *fsaverage6* surface mesh was used to map the x,y,z coordinates of the AHBA samples to the

closest vertices on the cortical surface prior to spatial interpolation via FreeSurfer reconstructions of the donor brain. In the condensed *Methods* section, we now refer the reader to the publication by Arnatkevičiūtė et al. (2019) for further details on the *abagen* toolbox. All subsequent preprocessing steps, including the use of *fsaverage6* for spatial interpolation, are detailed in the revised version of the manuscript.

- What are the 1670 samples mentioned in line 459? Samples of what? There are only 6 brains... are these spatial locations?

Author reply: In the context of the AHBA, ‘samples’ refer to specific locations or probes in the brain tissue from which gene expression data has been collected. These are also known as ‘well IDs’, which is a unique identifying for each probe acquired for particular donors. Each ‘well ID’ has a specific x,y,z coordinate that indicates its’ location. In total, the AHBA provides N=1670 samples across donors and hemispheres that are reliably measured in cortical brain tissue. We have included additional clarification on this point in the revised manuscript (see 1st paragraph of the *Methods* section).

- Why are coordinates only "roughly" symmetric (line 480). It should be possible to create perfectly symmetric templates by averaging coordinates in both sides. How "rough" is the residual asymmetry?

Author reply: We appreciate the opportunity to provide further details on the FreeSurfer symmetrical template (i.e., *fsaverage_sym*).

Projecting left (or right) hemisphere data onto the collateral hemisphere was performed using FreeSurfer ‘*xhemi*’ tools, a collection of routines developed to perform interhemispheric surface-based analysis published by Greve et al. (2013). Here, the data is originally projected onto the *fsaverage_sym* template, which is mathematically symmetrical, and has been constructed by averaging the cortical surfaces of multiple subjects. It is a modified version of the standard *fsaverage* template, which is not inherently symmetrical as the left and right hemispheres are averaged separately during template construction. Following registration to the *fsaverage_sym*, the data was backprojected onto the contralateral hemisphere of the *fsaverage6* prior to spatial interpolation. This necessarily leads to slight asymmetries given the inherent (i.e., nonsymmetrical) nature of the *fsaverage6*. Spatial interpolation was performed separately for each hemisphere, and the mRNA expression overlays are almost perfectly aligned across hemispheres. To demonstrate this point, we have now included an additional figure to the supplement, that shows the spatially interpolated mRNA data of some of the molecular targets examined in the current study for both hemispheres (see Extended Data Fig. 1). Moreover, we have provided further references to the FreeSurfer *xhemi* tools in the *Discussion* section.

- Why are there duplicate vertices? (line 482)

Author reply: For spatial permutation, each sample's x,y,z coordinates in volume space were projected onto the closest FreeSurfer *fsaverage6* vertex in surface space. In a few instances, samples from different donors were assigned to the same vertex. In such cases, the mRNA values at that vertex were averaged across donors. A similar approach was adopted by Gryglewski et al. (2018). However, since the data was previously normalized using the *abagen* toolbox, and this occurred in only a few instances, their impact is negligible. We have now added further clarification of our approach to the *Methods* section.

- Is it really necessary to do kriging to upsample data? If those 1670 samples represent spatial locations in the cortex, wouldn't it be more faithful to the data to downsample the in vivo imaging to that resolution, like the authors do for their second analysis that starts in line 563? It would also dispense with the need to do PCA and the need to deal with loading to interpret results, while still using BrainSMASH for inference.

Author reply: The reviewer raises an important point, which aligns with a comment from Reviewer #3. We therefore kindly direct the reviewer to our response to Reviewer #3's first comment on page 14, where we address this issue in detail.

- What does the \$k_{nn}\$ parameter mean? Why did it have to vary?

Author reply: In the context of variogram fitting, the 'knn' parameter refers to the number of nearest neighbors that are taken into account during the fitting procedure. It is therefore crucial for spatial interpolation, as it determines the local neighborhood (i.e., data points) used to estimate the spatial properties at unsampled locations based on the empirical variogram. Thus, different knn's can lead to vastly different variogram fits, which will significantly impact on the quality of the spatial prediction. The knn parameter can vary between 1 and the total number of vertices characterizing a cortical mesh (e.g., 41k for the *fsaverage6*). If chosen too small (or too large), a bad model fit will be obtained. In the current study, we therefore employed an optimization approach, which iteratively assessed the variogram fit based on the Normalized Root Mean Squared Error (NRMSE) across a number of knn values, ranging from 1,000 to 40,000. We then used the knn providing the best model fit for spatial interpolation. Notably, increasing the knn parameter significantly increased the computational load, especially for high-resolution cortical meshes such as the *fsaverage6*. For a single IDP, the optimization procedure can take anything between 3-5 hours using 10 CPUs. This is also the reason, why we precomputed the surrogate patterns for the 9 co-expression gradient maps prior to the statistical assessment of spatial correlations. This significantly improves computation time, and provides comparable results to utilizing N=1,000 permutations of the target pattern. We have now added further clarification of our approach to the *Methods* section (see section on

‘Transcriptomic decoding of surface-based IDPs using spatial autocorrelation-preserving null models’).

- Correction for multiple testing as described up to line 558 is ok. But maxT across PCs does not seem right, but in any case, very little details and no references are provided. What exactly was done here?

Author reply: We appreciate the reviewer’s question and have now extended the *Methods* section to clarify our approach.

Based on the N=1,000 surrogate maps for each of the nine PCs (i.e., gradient patterns), we derived two empirical distributions of null correlations: first, the distribution of null correlations within PCs that served as spatial null models for the mRNA expression pattern of each gene (also shown in Data Fig. 2c). This allowed us to derive a permutation *p*-value to assess the spatial correlation between the target IDP and the mRNA expression pattern of a particular gene, approximated by its respective gradient pattern. Second, according to the MaxT procedure, we derived the distribution of maximal spatial correlations across permuted PCs as a surrogate for individual genes (also see Data Fig. 2c), which allowed us to adjust the nominal permutation *p*-value with respect to the maximal spatial correlation observed across gradient patterns, and so to account for the large number of spatial null correlations examined across genes. However, the original MaxT procedure proposed by Westfall & Young (1993) has to be conducted across N=1,000 permutations of each gene or each target pattern, which was not possible given the high computational demands associated with generating the surrogate maps.

As part of the revisions, we have now repeated the analysis using 1,000 permutations of the target patterns instead of the gene expression profiles. This analysis demonstrates that (1) the permutations of the gradient patterns constitute valid spatial null models for the respective genes (i.e., genes with high loadings on a particular gradient pattern), and (2) the empirical null distribution across PCs approximates the empirical null distribution of spatial correlations across genes. These crucial additional analyses have now been included to the Supplementary Materials of the manuscript (see Extended Data Fig. 7 to 9), and so provide further evidence to support the validity of our gradient-based approach.

- If the method described between lines 562 and 577 does not take spatial autocorrelation into account, why use it? Why not just go straight with the GLS model?

Author reply: We much appreciate the reviewer’s question with regards to the models taking spatial autocorrelation into account (e.g., GLS, vertex-based approaches) vs. the LME approach, which is the only approach that does not account for spatial autocorrelations. Instead, the LME approach looks for consistencies in the correlations between IDP and gene expression patterns across donors. This approach was initially implemented to emulate the gene expression

decoding implemented within *Neurosynth* (<https://neurosynth.org>), an online platform for the automated synthesis of function MRI data (see Gorgolewski et al., 2014). So far, the *Neurosynth* decoding approach was the only approach not based on cortical parcellations, but instead transformed the surface-based data into standard stereotactic volume space (i.e., MNI305) by drawing spherical regions of interest at each AHBA sampling site. Thus, the surface-based IDP was downsampled to the resolution of the AHBA atlas. A linear mixed effect (LME) model was then fitted for each gene with donor as random effect variable. In this model, spatial autocorrelation is less of an issue, as model fitting occurs within donors, where there are fewer samples overall, and so the existing samples are located spatially further apart.

The *Neurosynth* decoding approach has also been utilized in several previous publications (e.g., Ecker et al. (2022)), and is currently no longer supported. Therefore, having alternative software implementations is advantageous as it allows us to compare results with previous findings. However, downsampling high-resolution IDPs to the native resolution of the AHBA has inherent limitations that were outlined in our reply to the 1st comment by *Reviewer #3* on page 14). Last, each decoding approach offers particular advantages and disadvantages. Some methods are highly conservative and specific in highlighting genes associated with an IDP but may be too stringent for exploratory analysis. Others are more sensitive but less specific, and making them better suited for exploratory purposes. Thus, it is important to implement a variety of techniques tailored to different scientific purposes. We have now added further justification of our approach to the *Methods* section, also with regards to determining the robustness of the vertex-level decoding compared to two alternative methods utilizing existing AHBA samples exclusively (i.e., without spatial interpolation).

- Why z-scoring cortical thickness was needed? The text says "to make individuals comparable" (lines 631-632) but that is too vague. Z-scoring on a per-vertex basis, done across the 279 subjects, surely alters the relative differences between neighboring vertices, and certainly impacts the spatial correlations with the expression maps. Shouldn't the raw cortical thickness values have been used instead when defining the clusters of subjects/strata?

Author reply: This is an important point and we thank the reviewer for asking us to further justify our approach to standardizing IDPs.

CT is influenced by various genetic and environmental factors during development and follows a similar pattern across individuals. A consistent pattern is therefore emerging that is dominated by large-scale CT trends or gradients across the cortex. These gradients are significantly influenced by age, biological sex, and other demographic measures, which causes individuals to be similar with respect to absolute CT variations. Given the consistency of these patterns, it is to be expected that transcriptomic decoding using absolute CT measures would yield a similar transcriptomic association or profile across individuals.

What is highly unique, however, is the neuroanatomical pattern by which individuals deviate from the typical trajectory of CT development. Such deviations from the predicted range are typically analysed within so-called normative modelling approaches, where a 'normative' (i.e.,

neurotypical) range of variation is initially established, providing a reference against which individual differences can be assessed. These studies, among others, demonstrate that an individual's neuroanatomy is characterized by highly personalized patterns of CT variability, which may function as a unique neuroanatomical "fingerprint" (e.g., Zabihi et al. (2020)). Furthermore, it appears that the extent to which individuals deviate from the typical trajectory of brain development, rather than absolute cortical thickness variations across the cortex, is associated with the prevalence of neuropsychiatric symptoms.

In the current study, we therefore examined deviations (i.e., residuals) from the typical trajectory rather than absolute CT measures to associate IDPs with the expression pattern of candidate genes, and to establish their association with symptoms of anxiety and depression. This is an important point, which is now further discussed in the *Discussion* section. Moreover, we have added further details to the *Methods* section (see 'Transcriptomic alignment between IDPs and GBA_AR gene expression patterns').

- In line 223 we learn about a tool the authors seem to have used but that isn't described in the Methods (Protein Atlas). Please, give details of the role of this tool in the Methods.

Author reply: We apologize for not providing further details on the Human Protein Atlas (HPA), which can be assessed at <https://www.proteinatlas.org>. Similar to the AHBA, the HPA is a comprehensive online resource that provides detailed information about the expression and localization of proteins in human tissues and cells. As such, it is more of a database than an analytical tool, which is also why we did not include further details on the atlas to the Methods section. However, further information on the HPA has now been added to the revised version of the manuscript.

C) Minor:

- The font used is too small even at size 11! Consider using the default fonts in MS Word to help us, poor-sighted reviewers! :-)

Author reply: We much apologize for the small font size. The font has now been changed to Times New Roman and a size of 12.

- Be judicious in the use of the word "prediction", here sometimes used to refer to the result of kriging, sometimes as the dependent variable in a statistical model. Currently, given the huge popularity of machine learning, the term "prediction" gained a new meaning that is not how it's used here.

Author reply: We apologize for not being precise enough in the wording of the original manuscript. In the context of machine learning, the term ‘prediction’ is typically used to refer to the estimated value for an unknown data point, such as one in a ‘test set’. This should be distinguished from in-sample estimates of associations obtained via correlation or regression analyses, which are obtained by minimizing the differences between the fitted and observed (i.e., known) data points, thus resulting from an optimization procedure. Based on this rationale, it is therefore accurate to utilize the term prediction in the context of spatial interpolation, as this involves the prediction of unknown mRNA expression values using existing AHBA samples. We have, however, revised the manuscript to remove the term ‘prediction’ in the context of the regression analysis presented in Data Fig. 6.

- Same for "decoding". It isn't clear anything is being "decoded" in the manuscript...

Author reply: In the context of gene expression, ‘decoding’ refers to the process of interpreting and analyzing patterns and levels of gene expression data in biological samples to gain insights into cellular functions, biological pathways, and mechanisms. In the context of imaging transcriptomics, the biological ‘sample’ is an imaging derived phenotype (IDP) characterized by a surface-based pattern of neuroanatomical variability in our case. The term ‘gene expression decoding’ has now been explicitly defined in the 2nd paragraph of the *Results* section.

- The paper hedges its bets a lot. Everything is "putative

" (the word appears 11 times). If you can say something, and have supportive evidence, please say so.

Author reply: In accordance with the reviewer’s suggestions, we have revised the manuscript to incorporate alternative phrasing wherever possible, and have eliminated several instances of 'putative'. We hope these changes have enhanced the readability of the manuscript and also mitigated ambiguities.

- The hierarchical clustering (starting from line 607) depends only on gene expression data from the AHBA, right? If so, consider swapping the order of presentation with the previous subsection on PET data (that starts from line 587). The reason is just for clarity, in case the hierarchical clustering of IDPs does not depend on the PET analysis (as it seems).

Author reply: We confirm that the hierarchical clustering of GABA_AR subunits based on mRNA expression patterns and the hierarchical clustering of IDPs were conducted independently of the PET data, although these analyses were motivated by the findings from the transcriptomic decoding of the PET atlas data. We have therefore structured the Results

section accordingly; i.e., to initially present the PET-based findings that served as an evaluation of the different decoding techniques. Based on these findings, we subsequently employed the clustering approaches to cluster GABA_AR subunits genes, and to cluster IDPs based on their differential association with subunit co-expression clusters. Therefore, the manuscript is structured to first present the methodological framework, followed by its application. As part of the revisions, we have now rephrased the *Introduction* to outline the study's rationale more clearly. We hope this offers a clearer framework that will help readers more easily navigate the structure of the results sections.

- "bifuricated" -> bifurcated (line 268)

Author reply: We apologize for this typo, which has now been correct.

All and all, I find the manuscript sound on its substance, and provides a valid, useful resource for people interested in using the Allen Brain data with in vivo imaging. But the paper is needlessly wordy, with many distractions that could (should in my opinion) be reduced.

Reviewer #5 (Remarks to the Author): Thank you for the opportunity to review your manuscript. I appreciate the chance to engage with your work and to provide feedback.

Linking gene expression data with imaging-derived phenotypes to explore functional mechanisms is a promising approach in imaging transcriptomics. I commend your validation against PET data and the subsequent analysis of cortical expression patterns of GABAA-receptor subunits. However, I believe that the statements about the association between GABAA-receptor subunits and specific behavioral traits need further clarification and thus a major revision of the manuscript. My main concern relates to the methodology used to elaborate on how distinct cortical expression patterns relate to specific behavioral symptoms:

- Even though the approach to guide targeted drug discovery by linking gene expression patterns to in-vivo symptoms is promising, statements such as "Yet, proof-of-concept that neurotransmission-related patterns of gene expression predict specific symptoms and traits, and so could inform targeted (i.e., behaviorally guided) drug discovery, remains missing." (lines 99-101) may be too far-fetched at this stage.

Author reply: We respectfully acknowledge the reviewer's perspective that this statement may be premature at this stage, as our findings are based on statistical associations rather than predictive models. As has also been noted by other reviewers, it will therefore be crucial in the future to explore the clinical utility of our approach in real-world clinical trials, i.e., to predict response to treatment on the case-level following GABA-ergic system modulation, or to explore other use cases in the clinical setting. While this goes beyond the scope of the present study, it is an important next step, which we are currently focussing on. In response to the reviewer's comment, we have revised the *Introduction* section to more accurately reflect our approach and findings. Additionally, in the *Discussion* section, we have emphasized the importance of validating and translating these findings into clinical applications as key directions for future research.

- The authors repeatedly write about transcriptomic characteristics of individuals, suggesting that data of single subjects were linked with behavioral traits (e.g., lines 111-114: "Taken together, our study suggests that individuals can be stratified based on their transcriptomic association with two regionally distinct GABA_AR subunit classes; one predominantly expressed in the limbic circuitry, and one with a regionally unspecific cortical expression landscape.") However, the analyses relate to the population-level as well as public atlas data, which limits the impact on "personalized interventions" or "personalized treatment strategies", which was stated in the introduction. To enhance the clarity of the manuscript, I recommend providing more detailed explanations of the underlying data basis. In particular, it should be stressed that the analyzed data sets are independent. In contrast, the information about the MR imaging data was provided rather late in the methods section. The authors should inform about

the measures of cortical thickness by relocation of the corresponding paragraphs to an earlier part of the manuscript.

Author reply: We appreciate the reviewer's comment that our study has not yet focused on potential clinical applications, such as the prediction of response to treatment in the context of personalized medicine approaches. As noted by the reviewer, this would require additional data acquired longitudinally in the same set of individuals (e.g., pre/post pharmacological intervention), in addition to out-of-sample validations. As mentioned above, this extends beyond the scope of the current study, but it is one of our team's primary research objectives in the future. To address the reviewer's comment, we are outlining the limitations of the current investigations more clearly in the *Discussion* section of the revised manuscript, and have also highlighted specific future research directions related to this point (see last paragraph of *Discussion* section).

Additionally, we have included information on the imaging phenotypes earlier in the manuscript (see last paragraph of the *Introduction*), also providing further justification for focusing on measures of cortical thickness in particular.

- In my opinion, the first two paragraphs of the results section (about "Spatially-dense (i.e., vertex-level) representations of the human brain transcriptome" and "Analytical framework for the transcriptomic decoding of high-resolution surface-based neuroimaging phenotypes") need shortening. There are too many details regarding the methodology that can be moved to the methods section.

Author reply: We have now significantly revised the *Results* section to remove redundancies, and condensing individual paragraphs in accordance with reviewers' suggestions. In particular, we have relocated a lot of detail from the *Results* to the *Methods* section, which helped to shorten the manuscript overall.

- In general, multiple paragraphs of the manuscript (in the introduction, results, and discussion sections) seem redundant and should be condensed accordingly.

Author reply: We greatly appreciate the reviewer's notion that the original manuscript may have been overly lengthy and contained too many redundancies. This was also brought to our attention by other reviewers, and we apologise for the lengthiness of the paper. We have now significantly shortened the manuscript by removing redundancies and condensing individual paragraphs in accordance with the reviewer's valuable suggestions. We believe that this has greatly improved the readability of the manuscript, and also narrowed down its focus. We are therefore grateful to the reviewers for giving us the opportunity for revising the manuscript.

- The authors state that they "generated a high-resolution surface representation of the AHBA human brain transcriptome following a similar approach as described by Gryglewski et al. (2018) 19". In the discussion sections, the authors should discuss in more detail why they consider their approach superior to other previously published approaches.

Author reply: A similar question was raised by Reviewers #1 and #2, who suggested that we compare our vertex-level mRNA expression patterns with the whole-brain mRNA patterns reported in the original study by Gryglewski et al. (2018). As mentioned above (see page 3), we consider the work by Gryglewski et al. (2018) absolutely vital to the field of imaging transcriptomics, as Gryglewski et al. were the first to provide gene expression profiles across the entire cortical surface, at a resolution that is comparable to conventional surface-based IDPs.

The main difference between our approach and that of Gryglewski et al. is in how we identified existing AHBA samples with a vertex's neighborhood, which formed the basis for the spatial interpolation. In the original publication by Gryglewski et al., spatial interpolation was based on the nearest samples in Euclidean space. Although this significantly enhances computational efficiency, it does not account for the complex geometric convolution (i.e., folding) of the cortical surface. Specifically, two vertices may appear close in Euclidean space but could be far apart when considering distances along the cortical surface. Drawing on our prior expertise in geodesic distance mapping (see Ecker et al., 2013), we identified AHBA samples for spatial interpolation based on the geodesic, rather than Euclidean, vertex neighborhood - meaning we used the shortest distance on the cortical surface rather than in space. Although this approach is computationally intensive (e.g., requiring approximately two weeks on 30 CPUs to spatially interpolate mRNA expression values for a single hemisphere), it should theoretically produce more accurate maps.

Overall, however, both approaches lead to highly correlated mRNA expression signatures, which we have now directly compared in the revised version of the manuscript (see Extended Data Fig. 2). More specifically, the spatial correlation coefficients between mRNA expression signatures exceed a value > 0.9 across approaches, which substantiates the validity and robustness of the presented mRNA signatures. Additionally, we have provided further clarification in the *Methods* and *Results* section regarding the differences between the Gryglewski approach and the methodology used in the present study.

- As a conclusion, the authors state: "Consequently, we demonstrated that individuals with delayed maturation in the limbic neurocircuitry, which was characterized by high expression of the $\alpha 2$ -containing GABAAR subunit Cluster 1, also had significantly higher levels of anxiety and depression than individuals falling into the $\alpha 1$ -containing GABAAR subunit Cluster 2, particularly during adulthood." This statement lacks clarity regarding the relationship between the analyzed (independent) data sets. Explaining the (statistical) association used to assume this (biological) relationship would enhance the comprehensibility and credibility of the findings.

Author reply: In this sentence, we aimed to convey that individuals with a higher degree of neuroanatomical diversity in brain regions with high levels of expression of genes within GABA_AR subunit *Cluster 1*, i.e., the ‘limbic’ cluster that also contained the alpha 2 subunit, also had higher levels of anxiety and depression, particularly during adulthood. Our findings thus support the critical role of the limbic neurocircuitry in mediating affective functions, and highlight the involvement of GABA_AR subunit *Cluster 1* specifically in their regulation. The conclusion have now been clarified in the revised *Discussion* section of the manuscript.

Further remarks:

- In the introduction the outdated wording "typical and atypical brain organization" should be updated.

Author reply: We much appreciate the reviewer’s suggestion to update the wording with regards to brain organization. After consultations with the consortium’s co-creation group, we are now referring to neurotypical and neurodiverse brain organization.

- The authors use the word "putative" several times in the introduction. I suggest refining the language by using alternative phrasings, avoiding the repetition of "putative".

Author reply: In accordance with the reviewer’s suggestions, we have revised the manuscript to incorporate alternative phrasing wherever possible, and have eliminated several instances of 'putative'.

- Lines 90-91: "For example, MR of GABA requires dedicated sequences, yet only report on ‘bulk’ GABA levels limiting interpretation 11." I believe, the authors wanted to write "MRS of GABA" ?

Author reply: Thank you for highlighting this typo to us. This has now been corrected.

- Lines 102-104: "In the current study, we therefore (i) present a novel approach for the transcriptomic decoding of high-resolution surface-based IDPs or targets patterns using spatially-dense representations of the human brain transcriptome generated from the AHBA." I believe, the authors wanted to write "target patterns" ?

Author reply: We would like to apologies for this typo, which has now been correct.

- Lines 115-117: "Based on these findings, we believe that the investigation of neurochemical pathways via fine-grained cartographic mapping of human gene expression offers promise for personalized medicine strategies, and represents a novel avenue to in vivo pharmacogenomics." I believe, the authors wanted to write "fine-grained" ?

Author reply: This has now been corrected.

- Lines 384-387: "Cortically, the expression of these subunits was tightly coupled with the regionally-unspecific expression pattern of genes in subunit cluster 2, suggesting that heteromeric GABAAR isoforms composed of $\alpha 1\beta 2\gamma 2$ subunits may play a crucial role in mediating a wide range of neurocognitive functions. " This sentence is hard to understand and I suggest another wording.

Author reply: We appreciate the reviewer's suggestion to rephrase the sentence for better clarity. We have now extensively revised the *Discussion* section and have amended this sentence accordingly.

We sincerely thank the reviewers for their time and effort in evaluating the revised version of our manuscript. We greatly appreciate the remaining comments and have addressed them in the latest revision. Specifically, we have shortened the Results section to emphasize the novel aspects of our study. Additionally, we have included an analysis comparing the empirical variograms of the gradient patterns with those of individual gene expression signatures to further validate our approach. Below, we provide our responses to each individual comment.

Reviewer #1 (Remarks to the Author):

While the authors have revised the manuscript to highlight and align more with previous research, the majority of the work presented in this manuscript has still been carried out before in previous papers (Gryglewski et al., Hansen et al. 2022, Norgaard et al. 2021, and Sequeira et al.), and therefore the novelty of this work is still limited, given the currently high priority in discussing these methods. Of novelty, is the section associating GABA patterns with CT/symptoms in an independent cohort, and the authors should therefore instead focus on these analyses and interpretations.

The work presented prior to this could be added to the methods section, with references to the previous work developing these methods, and then describing any significant changes e.g. geodesic distance on the surface compared to euclidean. This is also what the authors state to be the goal of the work, as mentioned in the response letter, page 2 "Instead, we argue that cortical gene expression patterns as revealed by the AHBA implicate specific neural systems (i.e., cortical patterns), which in turn map onto specific behavioural symptoms and traits.". This is the novel part, and the authors should focus on this instead of presenting what they refer to as 'new methods', but as indicated by the authors also result in 'While the resulting surface-based gene expression signatures are expected to be theoretically more accurate, as they account for the highly complex pattern of cortical folding, they were also highly correlated with the mRNA expression profiles published by Gryglewski et al. (2018). Additionally, the maps showed strong agreement with the spatially-smoothed dense expression patterns recently released by Wagstyl et al. (2023) on a lower-resolution (30k) surface template 19', from page 11 in the main manuscript, and 'Our finding also aligns with previous evidence suggesting that BZ binding sites can be categorized based on α subunit isoforms and their specific behavioral implications (reviewed in 13)' from page 14.

Author reply: We thank the reviewer for suggesting that we further emphasize the novel aspects of our study, particularly those related to the exploration of cortical networks informed by candidate gene expression patterns. As recommended by the reviewer, we have streamlined the Results section by relocating all methodological descriptions to the Methods section. More specifically, we have removed the section titled ‘*Transcriptomic decoding of surface-based IDPs using spatially-dense representations of the human brain transcriptome*’ from the results and provide a more detailed description of the analytical framework in the Methods section. The Results section now begins with a comparison of techniques, which we believe is important

to justify the use of the vertex-level approach for examining the GABAergic system in the subsequent analysis, and also provides insights into the advantages and disadvantages of the different approaches. In addition, we have moved Figure 3, which provides an overview of the LME and GLS approach to the Appendix. We left Figures 1 and 2 in the main manuscript to illustrate the overall workflow. We hope these revisions help readers better focus on the novel aspects of the study while clearly outlining and justifying the chosen analytical framework.

To my previous question #4 on taking into account the relative expression of each GABA subunit, the authors highlight in their answer my main concern namely differences in baseline expression between subunits, and large variation in expression across the cortex. However, it is still not clear from the manuscript and response from the authors how this was addressed, except stating that they focused on spatial correlations instead of covariances and there it was not relevant. But this does not alleviate the main concern of having different variances explained for the different subunits, and how this might impact these analyses and interpretations. For example, as shown by Norgaard et al. 2021 and Sequeira et al. 2019, the two most abundant α -subunits are $\alpha 1$ and $\alpha 2$ with 16% and 11% variance explained, respectively. However, the remaining α -subunits only explain between 2-5% variance across all brain regions.

Author reply: We apologize if the reviewer felt that this comment was not sufficiently addressed in the last revisions of the manuscript. The relative percentages of explained variance mentioned by the reviewer seem to reflect the proportional contribution of each subunit to the total mRNA expression in the brain across all subunits (represented as ‘V’ in Table 2 in Norgaard et al. 2021). These values align with the percentages assigned to each subunit in Fig. 1a of Sequeira et al. (2019). These percentages thus appear to primarily reflect baseline differences in total (i.e., whole-brain) mRNA expression between subunits, rather than the proportion of regional variation in total GABA_AR subunit mRNA expression explained by individual subunits. For instance, Fig. 1a (Sequeira et al. 2021) indicates that the global (i.e. whole-brain) mRNA expression level of the alpha 1 subunit seems eight times higher than the expression level of the alpha 4 subunit. In turn, this indicates a significantly higher abundance of mRNA transcripts for alpha 1 across the brain, rather than subunit protein abundance *per se*. This highlights the important need for standardizing mRNA expression values across genes to ensure that the mRNA data is comparable.

In the study by Sequeira et al., all data was hence expressed as % of each subunit expression to the total mRNA pool across the brain, which made it possible to compare regional variations in gene expression across subunit genes. In our study, we utilized the *abagen* toolbox to standardize mRNA expression values across genes and donors, thus ensuring that the mRNA expression levels of all genes have the same mean and standard deviation. To emphasize this important preprocessing step, additional details on the normalization procedure have now been incorporated into the Methods section. Moreover, we would like to emphasize that our analyses primarily focused on the spatial coherence between surface-based patterns, i.e., similarities in regional variations across the cortex. As these were quantified by means of spatial correlation

coefficients, these are by definition independent of baseline differences in gene expression. In response to the reviewer's comment, we have now added further clarification of our approach to the Methods section. Furthermore, we have expanded on this topic in the 7th paragraph of the Discussion section (see page 13).

Reviewer #1 (Remarks on code availability):

This reviewer appreciates that code has been shared along with the manuscript.

Reviewer #2 (Remarks to the Author):

I thank the authors for responding to the reviewer comments in detail. The authors have clarified that they are using surrogate nulls of gradients to assess statistical significance of gene-IDP correlations. This is statistically incorrect. If the null hypothesis is that the correlation between a gene and an IDP are due to spatial autocorrelation, then the null needs to preserve the spatial autocorrelation of either the gene or the IDP. Gradients are not necessarily closely correlated with the gene of interest, does not necessarily have the same spatial autocorrelation, and represents a fundamentally different thing than the gene expression map itself. I appreciate that the authors ran the analysis making nulls from the IDP, and that they show that false positive rates are variable, but ultimately using the gradient as a null is an indirect method for assessing a relationship when there is an available direct method (generating surrogates on the IDP, using spins/surrogates/moran nulls/eigenstrapping on either phenotype, limiting computational burden by parcellating, etc). This method should be revised to apply the correct statistical null model.

Outside of this issue, the authors use appropriate methodology, the revised manuscript is much more clear and focused, and the figures are well made.

Author reply: We appreciate the reviewer's comment with regards to the validity of the spatial null model. We fully agree that any valid null model must preserve the spatial autocorrelation of either the gene (in our case) or the IDP. If either spatial permutation or autocorrelation-preservation fails, the resulting model may be overly conservative, thus reducing its sensitivity, or overly lenient, leading to a high rate of false positives.

Our study comparing different approaches has, however, explicitly shown that the gradient-based null model outperforms alternative approaches in terms of their sensitivity and specificity. In fact, our study demonstrates that 'spins' null models, which perfectly preserve the spatial autocorrelation of the target pattern, are statistically too lenient in the context of gene expression decoding, resulting in poor differentiation between genes, and a high rate of false positives (see Extended Data Fig. 4-6). This was not the case for the gradient-based

approach, which was among the most conservative approaches tested, and which displayed a similar sensitivity and specificity to the model generating N=1,000 surrogates of the target IDP while being computationally more efficient. Thus, based on these performance metrics that were evaluated against true positives (i.e., known genetic target), it can be inferred that co-expression gradients may serve as valid surrogates for individual gene expression signatures, as they are neither overly lenient (resulting in a high number of false positives) nor overly conservative (failing to correctly identify target genes).

Furthermore, given the importance of the issue raised by the reviewer, we have now compared the empirical variograms of the nine co-expression gradients with those of individual gene's expression signatures for a random selection of genes. We observed that there is a strong alignment between variograms, indicating that the spatial autocorrelation inherent in gradient patterns and their surrogates closely mirrors the spatial dependence observed in individual gene expression signatures (see Extended Data Fig. 14). These additional findings further support the use of co-expression gradients as surrogates for individual gene expression signatures to derive an empirical distribution of spatial correlations under the null hypothesis. We believe these additional analyses provide important support for the validity of our approach and have now been incorporated into the Methods section.

Reviewer #3 (Remarks to the Author):

I would like to reiterate the positive feedback I provided in my previous review. The paper presents a compelling approach to linking transcriptomics with imaging phenotypes, and I appreciate the authors' efforts in addressing the methodological concerns raised earlier.

That said, while the revisions have improved many aspects of the paper, I remain somewhat unconvinced about the broad applicability of interpolating gene data to generate continuous transcriptomic maps for all types of targets. I recently encountered a paper (<https://pubmed.ncbi.nlm.nih.gov/33722642/>) that demonstrates how sensitive imaging associations can be when using different transcriptomics methods on the same data, which raises some questions about the generalizability of the approach presented here. However, for the systems of interest in this study—particularly GABA—the validation seems robust, and the methodology is clearly fit for purpose in this specific context.

A suggestion I have for further enhancing the paper is to consider providing access to other targets and genes beyond GABA and serotonin. This would allow for independent validation of the methodology and expand its applicability across different systems. Opening up this aspect would strengthen the paper's contribution to the field and potentially broaden its impact. Unfortunately I could not see this information in the shared code.

On a narrative level, the paper now reads more cohesively, and the findings are well-aligned with the overall conclusions. This work represents an exciting opportunity to advance our understanding of the connections between transcriptomic data and imaging phenotypes.

I have no further comments at this time.

Author reply: We much appreciate the reviewer's positive feedback on the revised manuscript. We agree that in order to assess the robustness and generalizability of imaging transcriptomics studies, it is absolutely vital to replicate their findings across different approaches, and to develop field-accepted standards for harmonizing preprocessing pipelines for both AHBA and neuroimaging data. A significant step towards this goal was the release of the *abagen* toolbox by Markello *et al.*, which also allows for the standardization of mRNA expression values across genes and donors. In this context, we considered it essential to implement and evaluate various decoding techniques, including approaches with and without spatial interpolation, as well as those utilizing different statistical methods (e.g., LME, GLS, etc.). This also made it possible to identify the particular strengths and limitations of each technique, which need to be taken into account when interpreting the results. Recognizing its importance, the reference suggested by the reviewer has now been incorporated into the manuscript to underscore this key point.

We also thank the reviewer for the comment with regards to making mRNA expression data of alternative molecular targets available. The spatially interpolated mRNA expression signatures for all genes are provided via the `fsdecode` decoding toolbox, which is freely available on GitHub (<https://github.com/christineecker/fsdecode>). This toolbox includes a genes-by-vertex matrix of normalized, spatially interpolated mRNA expression overlays, stored in the `lh.rh.mRNA.fsavg6.fwhm5.rda` data object. The expression pattern of individual genes can be visualized using the `fs_plot_gene()` function. Additional documentation and tutorials will also be provided following the publication of our manuscript, which we hope will strength the paper's contribution and broaden it's impact.

Reviewer #4 (Remarks to the Author):

I appreciate the authors responsiveness to the previous round of reviews. They have satisfactorily addressed all comments I had made.

Only one minor item: there is an important emphasis on speed, and I note that the extreme value method used for correction for multiple comparisons (here said to be similar to Westfall & Young, 1993) appears in fact remarkably similar to the "tail approximation" presented in <https://doi.org/10.1016/j.neuroimage.2016.05.068>. I think the latter reference is more informative about the procedure, which in itself seems correct to me.

Author reply: We would like to thank the reviewer for bringing this important reference to our attention. The approach described by Winker et al. is indeed closely aligned with the permutation testing approach employed in the present study, and we have now incorporated this reference into the Methods section.

Reviewer #4 (Remarks on code availability):

I applaud Nature journals for offering the possibility of reviewing code but a better option would be to allow a faster track for corrections or retractions when code errors are found after publication by users who really do try to use the code.

As a concrete example, Marek et al., Nature, 2022, has a coding mistake that invalidates its conclusions. The issue was flagged in a preprint by Italian researchers. Yet, somehow, the paper has received no expressions of concern, no corrections, and no "Matters Arising", despite the fact that Nature editors have been made aware of the problem.

That is something that needs to be fixed: allow fast track post-publication correction after code reviews by users worldwide. It is hard to rely on the already limited time available by a handful of reviewers who voluntarily review the manuscript itself.

Author reply: We much appreciate the reviewer's comment with regards to code availability, which we absolutely agree with. In this era of high-speed research, it is crucial that code is accessible for testing by users worldwide and that FAIR principles are implemented at all stages of development, from raw data processing to manuscript preparation. Furthermore, given the complexity of analytical designs these days, it is essential to have mechanisms in place that facilitate substitution and/or retraction, which publishers must treat with the utmost seriousness.

Reviewer #5 (Remarks to the Author):

Dear Authors,

thank you for submitting the revised manuscript "Transcriptomic decoding of surface-based imaging phenotypes and its potential for pharmacotranscriptomics".

I acknowledge the revisions made to the Introduction and Results sections, which now provide clearer and more accurate reflections of your approach and findings. The revised Discussion section now explains the importance of validating and translating these findings into clinical applications and highlights the necessity for future research.

Minor: I approve your revision after the initial feedback; however, I suggest to add a paragraph describing in more detail your decision to leave out subcortical regions from your analysis.

Author reply: We thank the reviewer for this suggestion. We have now included a short paragraph outlining the reasons for excluding subcortical regions from the analysis in the Discussion section (see 4th paragraph).